# UNSUPERVISED MODE DISCOVERY FOR FINE-TUNING MULTIMODAL ACTION DISTRIBUTIONS

## ABSTRACT

We address the problem of fine-tuning pre-trained generative policies with reinforcement learning while preserving the multimodality of the action distributions of such policies. Current methods for fine-tuning generative policies (e.g. diffusion policies) with reinforcement learning improve task performance but tend to collapse diverse behaviors into a single reward-maximizing mode. To overcome this, we propose MD-MAD, an unsupervised mode discovery framework that uncovers latent behaviors in generative policies, together with a mutual information metric to quantify multimodality. The discovered modes allow mutual information to be used as an intrinsic reward, regularizing reinforcement learning fine-tuning to improve success rates while maintaining diverse strategies. Experiments on robotic manipulation tasks demonstrate that our method consistently outperforms conventional fine-tuning, achieving high task success while preserving richer multimodal action distributions.

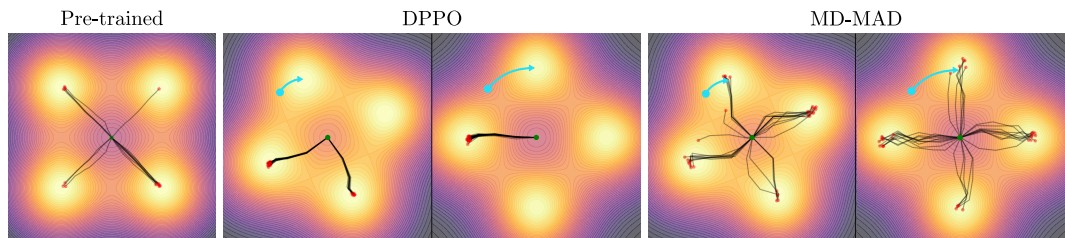

**Figure 1: MD–MAD for Preventing Mode Collapse**. Fine-tuning a pre-trained multimodal policy with standard RL often collapses its action distribution, eliminating modes discovered during pre-training. Our approach, MD–MAD, preserves multimodality while adapting the policy to the downstream task. In the figure, each panel overlays trajectories (black) starting from the origin in a reward landscape with four symmetric goals (bright peaks). *Left:* the pre-trained diffusion policy covers all four modes. *Middle (DPPO):* after RL fine-tuning under two rotated reward shifts (cyan arrows), trajectories collapse to a subset of goals. *Right (MD–MAD):* under the same shifts, the policy adapts without collapse and consistently recovers all modes.

## 1 INTRODUCTION

Robotic manipulation tasks are inherently multimodal, admitting diverse yet valid strategies: a cup can be grasped from either side, a block can be rotated clockwise or counterclockwise, and redundant kinematics allow the same goal to be reached via distinct motions. These scenarios naturally give rise to multimodal action distributions, whose preservation is key for policies that are robust, versatile, and adaptable to perturbations and unforeseen situations. Recent advances in generative policy learning have shown that expressive architectures such as diffusion (Chi et al., 2023; Kang et al., 2023; Psenka et al., 2023) and flow-based models (Lipman et al., 2022; Park et al., 2025) can capture such multimodality from demonstrations. However, their behavior is bounded by the coverage of the demonstration dataset. Reinforcement learning (RL) provides a natural mechanism to adapt and improve these pre-trained policies beyond demonstrations. Yet, RL fine-tuning often biases the policy toward few reward-maximizing behaviors at the expense of diversity, an issue that is further exacerbated when the fine-tuning reward is misaligned with the implicit objectives expressed in demonstrations (Zhou & Li, 2024; Brown et al., 2019). The central problem we address in this

work is therefore: *how can we fine-tune pre-trained generative policies with RL while preserving the multimodality acquired by supervised pre-training?*

Despite the community's growing interest in policies showcasing multimodal behaviors, little work systematically examines how RL adaptation affects multimodality. Existing research splits broadly into two directions. A first line of work focuses on fine-tuning expressive policies such as diffusion or flow models with RL to improve robustness and returns (Park et al., 2025; Ren et al., 2024; Chen et al., 2024). These approaches, however, do not account for multimodality in the action distribution, and often collapse the diverse behaviors captured during demonstration into a single dominant strategy. A second line of work begins to address multimodality more explicitly, for instance by proposing metrics to characterize it (Jia et al., 2024) or by leveraging language conditioning and instruction diversity (Black et al., 2024; Kim et al., 2024). Yet, these efforts either rely on assumptions that the number of modes is known in advance or that multimodality can be fully captured through language and labels. In practice, the modalities contained in the demonstration are usually unknown, and language provides only a coarse handle on behavior, which prevents precise encoding of low-level motor attributes such as magnitudes, scales, and endpoints (Lee et al., 2025).

In this work, we propose MD–MAD (***M**ode **D**iscovery for **M**ultimodal **A**ction **D**istributions*), a method to fine-tune expressive generative policies while explicitly preserving multimodality. We begin by introducing a measure of multimodality for this class of noise-conditioned generative models, such as diffusion and flow-based policies. Inspired by prior work on unsupervised skill discovery (Gregor et al., 2016; Eysenbach et al., 2018), we then design a mode discovery procedure that uncovers latent behavioral modes in pre-trained policies without assuming prior knowledge or relying on external annotations. This discovery process serves a dual purpose: it uncovers and makes controllable the latent modalities of the policy, and it enables the estimation of the policy's multimodality via a mutual information objective. This objective is subsequently employed as an intrinsic reward during reinforcement learning fine-tuning, regularizing the policy to retain diverse behaviors, as shown in Figure 1. We evaluated the proposed regularization method on multiple robotic manipulation tasks exhibiting multimodal behaviors. Across all tasks, our approach achieves comparable task success to standard fine-tuning while retaining action multimodality, demonstrating the effectiveness of our regularization objective.

In summary, our contributions are: **1)** A proxy to measure multimodality of generative policies that does not rely on mode labels or language supervision. **2)** An unsupervised mode-discovery framework enabling the identification of latent behavioral modes in pre-trained generative policies. **3)** A mode-preserving RL fine-tuning objective, where intrinsic rewards derived from discovered modes prevent collapse while improving task performance. **4)** An empirical evaluation on robotic manipulation tasks showing that our method preserves multimodality while enhancing task success.

## 2 RELATED WORK

We briefly review two areas closely connected to our central idea and contributions. For a more comprehensive discussion on related work, see Appendix A.

**Fine-tuning of Pre-trained Generative Policies.** Diffusion- and flow-based models provide expressive policy parameterizations for multimodal action distributions, but fine-tuning them with RL is challenging due to sequential sampling and the cost of backpropagating through the generative process. Recent work addresses these issues through three main strategies: direct fine-tuning, residual policies, and steering policies. *Direct fine-tuning* approaches adapt the network weights either by distilling the model into a one-step sampler for easier backpropagation (Park et al., 2025; Chen et al., 2024), by casting the denoising process as a sequential decision problem (Ren et al., 2024), or by using differentiable approximations that allow offline Q-learning without backpropagating through all denoising steps (Kang et al., 2023). *Residual policy* learning methods instead freeze a pre-trained generative policy and learn a small corrective controller via RL to address execution errors (Ankile et al., 2024; Yuan et al., 2024). These techniques can yield substantial performance gains over pure imitation learning (IL), and potentially preserve the diversity learned from demonstrations. *Steering policy* methods instead bias the sampling process toward high-value actions without modifying the generative model itself Wagenmaker et al. (2025); Yang et al. (2023); Wang et al. (2022). A common limitation of the aforementioned approaches is that they lack explicit mechanisms to preserve

multimodality, and often converge to a single reward-maximizing solution. Our work extends the steering-policy framework of Wagenmaker et al. (2025), by introducing mode discovery to discover and control latent modalities while biasing all behaviors towards higher rewards.

**Skill Discovery.** Multimodal behavior learning has also been studied through unsupervised skill discovery, which aims to acquire diverse and distinguishable behaviors without external rewards. A common approach is to maximize mutual information between a latent skill variable and visited states or trajectories (Gregor et al., 2016; Eysenbach et al., 2018). Most existing methods train policies from scratch in reward-free settings, but diversity alone often leads to skills that may be ill suited for downstream tasks. To address this, prior work has incorporated language guidance (Rho et al., 2025), extrinsic rewards (Emukpere et al., 2024), or state-space regularization (Park et al., 2023). Our approach differs by leveraging a pre-trained generative model to uncover useful behaviors already encoded in demonstrations. To our knowledge, we are the first to study skill discovery in this context, treating skills as modes in the latent noise space of a pre-trained generative policy.

## 3 PROBLEM FORMULATION

Formally, we study the problem of fine-tuning a pre-trained diffusion policy using reinforcement learning to maximize expected return, while explicitly preserving the multimodality of the action distribution induced by demonstrations. Specifically, we consider multimodality that may arise either from heterogeneity in task goals or from the existence of multiple feasible trajectories leading to the same goal. We model the environment as a Markov Decision Process (MDP) described as a tuple $(\mathcal{S}, \mathcal{A}, r, p, \gamma)$, with state space $\mathcal{S}$, action space $\mathcal{A}$, reward function $r$, transition dynamics $p$, and discount factor $\gamma \in [0, 1)$. The objective of RL is to learn a policy $\pi_\theta(a \mid s)$ maximizing the expected discounted return

$$J(\pi) = \mathbb{E}_\pi \left[ \sum_{t=0}^\infty \gamma^t r(s_t, a_t) \right],$$

where $s_t$ and $a_t$ are distributed according to the transition dynamics $p$ and the policy $\pi$.

**Pre-Trained Multimodal Generative Policies.** We assume access to an offline demonstration dataset $\mathcal{D} = \{\tau^{(i)}\}_{i=1}^N, \tau^{(i)} = (s_0^{(i)}, a_0^{(i)}, \ldots, s_{T_i}^{(i)}, a_{T_i}^{(i)})$, collected from diverse behavioral policies (e.g. human demonstrations). We pre-train a generative policy $\pi_\theta$ on $\mathcal{D}$ via imitation learning, and we assume that the trajectory-level diversity in the dataset induces multimodality in the action distribution, similarly to Chi et al. (2023). When relevant, we make explicit the dependence of the generative policy on its input noise variable $w \in \mathcal{W}$ by denoting it as $\pi_\theta(a \mid s, w)$. We model the modes of the policy $\pi_\theta$ using a discrete latent variable $z \in \mathcal{Z}$. Each value $z$ selects a particular instantiation of the policy inducing the trajectory distribution $p^\pi(\tau \mid z) = p(s_0) \prod_{t=0}^{T-1} \pi(a_t \mid s_t, z) \, p(s_{t+1} \mid s_t, a_t)$. Different values of $z$ therefore correspond to distinct self-consistent strategies present in the dataset, i.e., different behavioral modes. We assume the original modes $z \in \mathcal{Z}$ contained in the datasets are unknown but implicitly encoded in the pre-trained multimodal policy.

**Steerability Assumption.** We assume that the pre-trained generative policy $\pi_\theta(a \mid s, w)$ is *steerable*, in the sense that its behavior can be systematically influenced through the choice of the latent noise input $w \in \mathcal{W}$. A *steering policy* $\pi_\psi^\mathcal{W}(w \mid s)$, parameterized by $\psi$, selects which point $w$ in the latent-noise space to denoise, biasing the generative model toward different behavioral modes (Wagenmaker et al., 2025).

**Fine-tuning Objective.** Our goal is to fine-tune the policy $\pi_\theta$ in order to (i) maximize expected return and (ii) preserve the multimodality present in the pre-trained policy $\pi_\theta$. We formalize this as the regularized optimization problem

$$\max_\theta \ J(\pi_\theta) + \lambda \, \mathcal{M}(\pi_\theta),$$

where $\mathcal{M}$ denotes a multimodality measure of the generative policy, and $\lambda \geq 0$ balances task performance with diversity preservation. Importantly, we do not assume prior knowledge of the number of modes in $\pi_\theta$. Designing a practical measure for multimodality under these constraints is a central contribution of this work.

## 4 MODE DISCOVERY FOR RL FINETUNING

To fine-tune pre-trained diffusion policies while preserving multimodality, our method hinges on identifying and controlling latent behavioral modes of the pre-trained policy, which in turn enables us to regularize RL fine-tuning with a trajectory-diversity objective. To this end, our framework builds on three components: (i) we first introduce a tractable proxy to measure multimodality $\mathcal{M}(\cdot)$ in pre-trained generative policies based on mutual information; (ii) we then develop an unsupervised mode-discovery procedure by reparameterizing a steering policy $\pi_\psi^{\mathcal{W}}(w \mid s)$ through a latent variable $z \in \mathcal{Z}$, enabling us to uncover and control the behavioral modes of the pre-trained policy during training, while also providing an estimate of multimodality through mutual information. (iii) Finally, we use this estimate to construct a mutual information–based intrinsic reward and combine it with task rewards, regularizing reinforcement learning fine-tuning to improve task performance while explicitly retaining diverse behaviors. In what follows, we describe each component in detail.

### 4.1 MEASURING MULTIMODALITY IN GENERATIVE POLICIES

Classical definitions of multimodality characterize modes as local maxima of the explicit action distribution $\pi_\theta(a \mid s)$ (Stoepker & van den Heuvel, 2016), but this view becomes impractical for diffusion or flow-based policies, whose action densities are not available in closed form. To obtain a tractable surrogate, we make explicit the dependence of the policy on an input noise variable $w \in \mathcal{W}$ and write $\pi_\theta(a \mid s, w)$ with $w \sim \mathcal{W}$. We assume that the multimodality of the pre-trained policy $\pi_\theta(a \mid s)$ is realized through this latent noise, in the sense that there exists a set of states of non-zero measure for which different values of $w$ induce different action distributions, i.e., $\pi_\theta(\cdot \mid s, w_1) \neq \pi_\theta(\cdot \mid s, w_2)$ for some $w_1 \neq w_2$. Under this assumption, the latent $W$ and the action $A$ are statistically dependent given $S$, and therefore the conditional mutual information is strictly positive (proof in Appendix C)

$$I(W; A \mid S) = \mathbb{E}_{s \sim p(s)}\big[D_{\mathrm{KL}}\big(\pi_\theta(a \mid s, w) \,\|\, p(a \mid s)\big)\big] > 0,$$

where $p(a \mid s) = \mathbb{E}_{w \sim \mathcal{W}}[\pi_\theta(a \mid s, w)]$ is the marginal action distribution. Although $I(W; A \mid S) > 0$ is a valid proxy for multimodality in the pre-trained action distribution, this does not guarantee multimodality in the induced trajectories during fine-tuning, as action multimodality does not necessarily translate into trajectory multimodality. For example, in a kinematically redundant manipulator, multiple distinct joint-space actions can yield nearly identical end-effector motions, collapsing multimodality in the action distribution to a single mode in trajectory space.

Drawing inspiration from the unsupervised skill discovery literature, we address this problem by quantifying multimodality through a trajectory-diversity measure. In particular, rather than measuring multimodality at the action level, we consider the mutual information between the latent noise and the visited states, $I(W; S)$, which directly captures diversity in the induced trajectories (Gregor et al., 2016; Eysenbach et al., 2018; Sharma et al., 2019). Unlike skill-discovery methods, whose goal is to *learn* a latent space of skills from scratch, our setting begins with a pre-trained generative policy whose latent noise space $\mathcal{W}$ already encodes multiple behavioral modes. However, the corresponding modes are implicit in the structure of $\mathcal{W}$, which varies across time. In the next section, we introduce a steering policy $\pi_\psi^{\mathcal{W}}$ to uncover and control the latent behavioral modes in pre-trained policies, effectively lifting the representation from step-wise noise to trajectory-level structure.

### 4.2 MODE DISCOVERY OF PRE-TRAINED GENERATIVE POLICIES

Directly optimizing $I(W; S)$ is impractical as the implicit structure of $\mathcal{W}$ varies at every time-step and for each action-chunk, whereas the multimodal behaviors we are interested in emerge at the trajectory level. To overcome this problem and simultaneously obtain a structured and controllable representation of the policy modes, we introduce MD-MAD (*Mode Discovery for Multimodal Action Distributions*), which reparameterizes a steering policy with a latent variable $z \in \mathcal{Z}$ that organizes $\mathcal{W}$ into trajectory-level modes.

**Latent Reparameterization.** Let $\pi_\psi^{\mathcal{W}}(w \mid s)$ denote a steering policy that selects the latent noise $w \in \mathcal{W}$ seeding the denoising process. We introduce a latent variable $z \in \mathcal{Z}$ and define a latent-

**Figure 2: Unsupervised Mode Discovery via Latent Reparameterization of a Steering Policy.** An inference model $q_\phi(z \mid s)$ and a steering policy $\pi_\psi^{\mathcal{W}}(w \mid s, z)$ are trained jointly to uncover latent modes $z \in \mathcal{Z}$ in the frozen diffusion actor $\pi_\theta(a \mid s, w)$. The steering policy structures the noise space $\mathcal{W}$ according to $z$, inducing diverse actions $a \in \mathcal{A}$, while the inference model recovers $z$ to provide a variational estimate of $I(Z; S)$ (Eq. 2), used as an intrinsic reward during mode discovery and fine-tuning.

conditioned steering policy $\pi_\psi^{\mathcal{W}}(w \mid s, z)$, which induces the family of action distributions

$$\pi_{\theta,\psi}(a \mid s, z) \;=\; \int \pi_\theta(a \mid s, w)\, \pi_\psi^{\mathcal{W}}(w \mid s, z)\, dw. \tag{1}$$

Under this reparameterization, multimodality, or trajectory-level diversity, can be measured by the mutual information $I(Z; S)$, which places us back in the standard skill-discovery setting and allows us to leverage this class of methods to optimize the steering policy. By training the steering policy to maximize $I(Z; S)$, we encourage different values of $z$ to induce distinct state–trajectory distributions by steering the sampling of $\mathcal{W}$, thereby uncovering the modes implicitly encoded in the latent noise space. Distinct values of $z$ can therefore select different behaviors (modes) encoded by the *fixed* pretrained policy $\pi_\theta(a \mid s, w)$ through the steering policy $\pi_\psi^{\mathcal{W}}(w \mid s, z)$.

**Variational Lower Bound.** To optimize $\pi_\psi^{\mathcal{W}}$ via $I(Z; S)$ we follow standard practice in skill discovery (Eysenbach et al., 2018), which derives a variational lower bound on the mutual information by introducing an inference model $q_\phi(z \mid s)$ that approximates the posterior over latent codes. This yields

$$I(Z; S) = \mathbb{E}_{p(s,z)}\left[\log \frac{p(z \mid s)}{p(z)}\right] \geq \mathbb{E}_{p(s,z)}\left[\log q_\phi(z \mid s) - \log p(z)\right], \tag{2}$$

where $p(s, z)$ denotes the joint distribution induced by sampling $z \sim p(z)$ and rolling out a policy $\pi(a \mid s, z)$ in the environment. We refer to Eysenbach et al. (2018) for the derivation of Equation 2.

The log-posterior likelihood is used as an intrinsic reward for training the steering policy $\pi_\psi^{\mathcal{W}}$, aligning the RL objective with the identifiability of $z$. This establishes a feedback loop in which $q_\phi$ improves at classifying latent codes while the $\pi_\psi^{\mathcal{W}}$ is incentivized to select $w \in \mathcal{W}$ so as to induce trajectories that are consistent and discriminable. An overview of the method for mode discovery is illustrated in Figure 2.

### 4.3 POLICY FINE-TUNING WITH INTRINSIC REWARD

Recall from Section 3 that we formulated fine-tuning as maximizing task return regularized by a multimodality measure $\mathcal{M}$. Building on this, the variational lower bound introduced above provides a tractable instantiation of $\mathcal{M}$, which is leveraged as an intrinsic signal to preserve multimodality during fine-tuning. Concretely, we define the augmented reward

$$r_{\text{total}}(s, z) = r_{\text{env}}(s, a) + \lambda\Big( \log q_\phi(z \mid s) - \log p(z) \Big), \tag{3}$$

where $r_{\text{env}}$ is the environment reward and $\lambda \geq 0$ balances task performance with multimodality preservation. Directly combining task and intrinsic rewards may lead to premature collapse if the task signal dominates before the multimodal structure is discovered. We therefore adopt a two-stage scheme: first, the steering policy is trained with the intrinsic objective alone to uncover the modes of the pretrained policy; then the environment reward is introduced to guide fine-tuning toward high-return behaviors without destroying diversity. During mode discovery, we also apply a short-to-long horizon curriculum to stabilize learning. Algorithm 1 summarizes the overall procedure,

---

**Algorithm 1** Mode Discovery and Fine-Tuning of Generative Policies

---

1: **Inputs:** pre-trained diffusion policy $\pi_\theta(a \mid s, w)$; steering policy $\pi_\psi^\mathcal{W}(w \mid s, z)$; inference model $q_\phi(z \mid s, a)$; critic $V_\omega(s, z)$; latent prior $p(z)$, epochs $E$, episodes $N$, warm-up epochs $E_{wp}$; horizon scheduler $H_{schedule}(e)$

2: **Init:** $\psi, \phi, \omega$; set $\lambda \geq 0$

3: **for** $e = 1$ **to** $E$ **do**                                                                                                 ▷ epochs
4:    **for** $n = 1$ **to** $N$ **do**                                                                       ▷ episodes per epoch
5:       $H \leftarrow H_{schedule}(e)$                                                  ▷ curriculum horizon
6:       Sample $z \sim p(z)$ and rollout the policy:
7:         $w_t \sim \pi_\psi^\mathcal{W}(w \mid s_t, z), \quad a_t \sim \pi_\theta(a \mid s_t, w_t), \quad s_{t+1} \sim p(\cdot \mid s_t, a_t)$
8:       Intrinsic reward: $r_t^{int} \leftarrow \lambda\big(\log q_\phi(z \mid s_{t+1}) - \log p(z)\big)$
9:       **if** $e < E_{wp}$ **then**
10:          Policy reward: $r_t^{tot} \leftarrow r_t^{int}$                                ▷ Mode Discovery
11:       **else**
12:          Policy reward: $r_t^{tot} \leftarrow r_{env}(s_t, a_t) + r_t^{int}$        ▷ Policy Fine-tuning
13:       **end if**
14:    **end for**
15:    Update steering policy and critic using $r_t^{tot}$ (PPO):   $\min_{\psi,\omega} L_\pi^{PPO}(\psi) + c_V L_V(\omega) + c_\mathcal{H} L_\mathcal{H}(\psi)$
16:    Update inference model:   $\min_\phi L_q(\phi) = -\mathbb{E}\big[\log q_\phi(z \mid s)\big]$
17: **end for**

---

which is thoroughly described in Appendix D.3. While we adopt PPO (Schulman et al., 2017) in our experiments, our framework is agnostic to the specific RL algorithm used.

**Broader Use of the Framework.** While the formulation above fine-tunes the generative model indirectly via the steering policy, the framework is not limited to this case. Because the steering head actively explores diverse input-noise regions while pursuing reward, it can be combined with direct fine-tuning of the diffusion weights, acting as a structured exploration agent. At test time, the steering policy can either be retained—allowing explicit control over the behavioral mode—or removed, reverting to random sampling from the noise prior. Furthermore, while outside the present study, the learned latent space $\mathcal{Z}$ provides a natural basis for grounding semantic labels (e.g. language instructions) when limited annotations are available.

## 5 EXPERIMENTS

Our experimental evaluation is centered around three main questions: (i) Is the mutual information in Eq. 2 a valid measure of multimodality? (ii) Do existing fine-tuning techniques preserve multimodality? (iii) Does our method retain multimodality without sacrificing task performance? To answer these questions, we evaluated our method across a progressively more challenging set of environments: an illustrative 2D Gaussian-mixture reward landscape, multimodal manipulation tasks from ManiSkill (Tao et al., 2024) and D3IL (Jia et al., 2024), and additional high-dimensional and sequential domains including ANYmal locomotion and Franka Kitchen (Fu et al., 2020). We finally included ablations on key design choices.

**Baselines.** We benchmark our approach against representative strategies for on-policy fine-tuning of generative policies, focusing on diffusion models but noting that analogous evaluations apply to flow-matching policies. We include `DPPO` (Ren et al., 2024), as a direct finetuning approach, Policy Decorator (Yuan et al., 2024) as a residual fine-tuning approach (`RES`), and we consider Wagenmaker et al. (2025) as a steering policy `SP` based approach. For `DPPO` we select DDIM parameterization that reduces stochasticity while balancing $\eta > 0$ and the number of diffusion steps for stable weight updates. We further include a DDPM-based version that samples with the full denoising chain and fine-tunes the last 10 diffusion steps for completeness (`DPPO[10]`). Importantly, our approach is orthogonal to these categories and can be combined with any of them. Therefore, we report results both for the standalone baselines and their variants augmented with our multimodality regularizer, denoted as `X[MD-MAD]`, where `X` indicates the corresponding baseline. Full implementation details for all baselines and their regularized variants are provided in Appendix E.

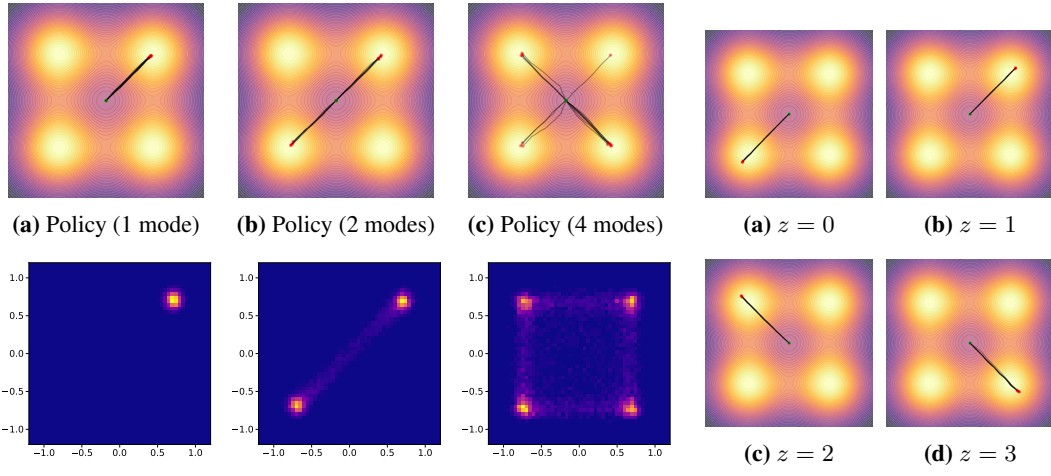

**(a)** Policy (1 mode)  **(b)** Policy (2 modes)  **(c)** Policy (4 modes)   **(a)** $z = 0$  **(b)** $z = 1$

**(d)** 1 mode  **(e)** 2 modes.  **(f)** 4 modes.   **(c)** $z = 2$  **(d)** $z = 3$

**Figure 4:** (Top) Trajectories generated from policies pre-trained. (Bottom) Monte Carlo estimate of the action distribution $(\Delta x, \Delta y)$ at $t = 0$.

**Figure 5:** Rollouts generated by steering the policy with latent codes $z \in \{0, 1, 2, 3\}$.

**Evaluation Metrics.** We assume access to the ground truth modes of the trajectories executed by the policy in simulation, and we evaluate fine-tuned policies along two axes: *task success* and *behavioral diversity*. We report overall success rate SR, and two mode-aggregated measures of the success rate to integrate behavioral diversity: the success rate weighted for each mode $\mathrm{SR_M} = \frac{1}{K} \sum_{i=1}^{K} \mathrm{SR}_i$, which guards against degenerate solutions (e.g., $100\%$ success on a single mode but failure on others), and mode coverage $\mathrm{mc}@\tau = \frac{1}{K} \sum_{i=1}^{K} \mathbf{1}\{\mathrm{SR}_i \geq \tau\}$, the fraction of modes solved above threshold $\tau = 0.8$. We further compute the entropy of the empirical distribution over modes among all rollouts: $H(\pi) = -\sum_{i=1}^{K} p_i \log p_i$, where $p_i$ is the fraction of episodes in mode $i$. All metrics are computed from $N = 1024$ evaluation episodes with fixed seeds for fair comparison, and we report both the mean and standard deviation over three independent runs with different random seeds.

## 5.1 2D GAUSSIAN MIXTURE

To study the proposed questions in a controlled setting, we designed a 2D navigation environment where the reward landscape is a mixture of $4$ Gaussians centered at fixed goal locations. We study both a *balanced* variant, where all goals have equal weight, and an *unbalanced* variant, where mode weights are randomized and normalized via a softmax, producing uneven but non-degenerate reward magnitudes. Further details and illustrations are provided in Appendix G.1.

**Mutual Information as a Proxy for Multimodality and Mode Discovery.** We first evaluate if mutual information provides a reliable proxy for multimodality. To this end, we construct expert datasets in the Gaussian-mixture environment containing one, two, or four goal modes, and train separate policies on each dataset (demonstrations are shown in Figure 10). Figure 4 shows rollouts of policies trained on each dataset (top row) alongside Monte Carlo estimates of their action distributions at $t = 0$ (bottom row). We hypothesize that a valid multimodality metric $\mathcal{M}$ should increase with the number of modes. To test this, we estimate $\mathcal{M}$ with Equation 2 by jointly training a steering policy and an inference model $q_\phi$ over a discrete latent space $\mathcal{Z} = \{0, 1, 2, 3\}$.

Table 1 reports the estimated mutual information and inference-model loss from $q_\phi$ on 1024 trajectories with randomly sampled $z \in \mathcal{Z}$. As expected, mutual information increases with the number of modes, while the loss decreases, indicating that $q_\phi$ reliably recovers latent codes when multimodality exists, but struggles in the unimodal case. These results support mutual information as a proxy for multimodality and as a useful training signal. Figure 5 further

**Table 1:** Mutual information and inference model loss.

| Policy | $\mathcal{M}$ | $q_\phi$ **Loss** |
|---|---|---|
| 1 mode | 0.00±0.00 | 1.38±0.00 |
| 2 modes | 0.58±0.02 | 0.82±0.02 |
| 4 modes | 1.06±0.00 | 0.33±0.02 |

**Table 2:** Evaluation on the Gaussian-mixture environment under two fine-tuning reward landscapes, and their unbalanced version (Unb.).

| Method | Goal [1] | | | | | Goal [2] | | | | |
|---|---|---|---|---|---|---|---|---|---|---|
| | SR (↑) | $SR_M$ (↑) | mc@80 (↑) | $\mathcal{H}$ (↑) | $SR_M$ (Unb.) (↑) | SR (↑) | $SR_M$ (↑) | mc@80 (↑) | $\mathcal{H}$ (↑) | $SR_M$(Unb.) (↑) |
| RES | $0.98_{\pm 0.02}$ | $0.98_{\pm 0.02}$ | $4.00/4$ | $1.00_{\pm 0.00}$ | $0.59_{\pm 0.07}$ | $0.92_{\pm 0.12}$ | $0.50_{\pm 0.00}$ | $2.00/4$ | $0.59_{\pm 0.13}$ | $0.50_{\pm 0.00}$ |
| SP | $1.00_{\pm 0.00}$ | $0.25_{\pm 0.00}$ | $1.00/4$ | $0.00_{\pm 0.00}$ | $0.17_{\pm 0.12}$ | $0.33_{\pm 0.47}$ | $0.08_{\pm 0.12}$ | $0.33/4$ | $0.00_{\pm 0.00}$ | $0.00_{\pm 0.00}$ |
| DPPO | $1.00_{\pm 0.00}$ | $0.58_{\pm 0.12}$ | $2.33/4$ | $0.40_{\pm 0.03}$ | $0.25_{\pm 0.00}$ | $1.00_{\pm 0.00}$ | $0.42_{\pm 0.12}$ | $1.67/4$ | $0.02_{\pm 0.02}$ | $0.00_{\pm 0.00}$ |
| DPPO[10] | $0.66_{\pm 0.32}$ | $0.16_{\pm 0.08}$ | $0.33/4$ | $0.00_{\pm 0.00}$ | $0.25_{\pm 0.00}$ | $0.32_{\pm 0.22}$ | $0.11_{\pm 0.05}$ | $0.00/4$ | $0.60_{\pm 0.22}$ | $0.14_{\pm 0.20}$ |
| RES[MD-MAD] | $1.00_{\pm 0.00}$ | $1.00_{\pm 0.00}$ | $4.00/4$ | $0.99_{\pm 0.00}$ | $1.00_{\pm 0.00}$ | $1.00_{\pm 0.00}$ | $1.00_{\pm 0.00}$ | $4.00/4$ | $0.94_{\pm 0.00}$ | $1.00_{\pm 0.00}$ |
| SP[MD-MAD] | $0.33_{\pm 0.47}$ | $0.33_{\pm 0.47}$ | $1.33/4$ | $0.46_{\pm 0.41}$ | $0.17_{\pm 0.12}$ | $0.33_{\pm 0.04}$ | $0.08_{\pm 0.12}$ | $0.33/4$ | $0.84_{\pm 0.14}$ | $0.03_{\pm 0.02}$ |
| DPPO[MD-MAD] | $1.00_{\pm 0.00}$ | $1.00_{\pm 0.00}$ | $4.00/4$ | $0.99_{\pm 0.00}$ | $1.00_{\pm 0.00}$ | $1.00_{\pm 0.00}$ | $0.75_{\pm 0.00}$ | $3.00/4$ | $0.74_{\pm 0.00}$ | $0.75_{\pm 0.00}$ |

shows that conditioning the steering policy on individual $z$ produces distinct, coherent trajectories, confirming that the latent space organizes noise into meaningful behavioral modes.

**Multimodality and Task Performance under Fine-Tuning.** We evaluate the performance of existing fine-tuning methods against our proposed MD-MAD in preserving multimodality when the reward used for adaptation differs from the one implicitly encoded in the demonstrations. To simulate this mismatch, we define two shifted reward landscapes obtained by rotating the Gaussian peaks used for demonstrations by $\frac{\pi}{8}$ and $\frac{\pi}{4}$, denoted as **Goal[1]** and **Goal[2]**. For each, we consider both a *balanced* variant, where all modes contribute equally, and an *unbalanced* variant, where the Gaussian weights are rescaled to produce asymmetric rewards (more details in Appendix G.1).

Table 2 reports results for both goals. Among baselines, the residual policy (RES) performs best, solving **Goal[1]** and retaining two modes in **Goal[2]**, as constrained corrections help preserve multimodality. Diffusion-based methods (DPPO, DPPO[10]) improve task success, with DPPO aided by extra denoising steps, but both collapse to fewer modes when rewards diverge from demonstrations. Steering alone (SP) is least effective, with limited success in **Goal[1]** and full collapse in **Goal[2]**. Multimodality retention further degrades in the unbalanced setting, where reward bias toward specific goals causes even strong baselines to collapse to dominant peaks. These results suggest that standard fine-tuning techniques fail to fully preserve the original multimodality as the reward landscape deviates from the distribution underlying the demonstrations.

In contrast, the [MD-MAD] variants preserve diversity more consistently: RES recovers full mode coverage across both goals and their unbalanced versions, and DPPO shows similar gains. For the SP method, [MD-MAD] mitigates but does not prevent collapse, indicating that additional fine-tuning of the original policy is required in this case. Overall, MD-MAD stabilizes fine-tuning in symmetric tasks and counteracts reward asymmetries that bias baselines toward fewer behaviors. Qualitative visualizations of the trajectories learned by the DPPO and RES[MD-MAD] policies are shown in Figure 1, while Appendix G.3 reports ablations on the dimensionality of $\mathcal{Z}$.

## 5.2 ROBOTIC MANIPULATION

Next, we evaluate our method on three simulated robotic tasks: *Reach*, *Lift*, and *Avoid*, implemented on ManiSkill (Tao et al., 2024) and visualized in Figure 12. Multimodality arises either from goal diversity or trajectory diversity in achieving the same goal. For each task, we collect 1000 demos with a motion planner and pre-train a diffusion model for 1000 epochs. Subsequently, dense or intermediate rewards are provided to support fine-tuning, and a heuristic is used to assign trajectories to modes for evaluation. Further implementation details are given in Appendix H.

**Standard Fine-tuning.** Table 3 summarizes results for baselines without explicit multimodality preservation. In *Reach*, all methods fine-tune the pre-trained policy without collapse, indicating that the inherent exploration of diffusion policies suffices to adapt both modes. In *Lift*, fine-tuning improves success rates but fails to consistently solve both modalities; only the steering-based baseline (SP) maintains higher entropy, showing that KL regularization with a Gaussian prior on the output of the steering policy (to enforce closeness with the original input noise distribution) can partly mitigate collapse, albeit at the cost of performance. In *Avoid*, fine-tuning achieves high task success but eliminates multimodality, driven by (i) reward mismatch between pre-training and fine-tuning,

**Table 3:** Baselines fine-tuning.

| Method | SR(↑) | SR$_M$ (↑) | mc@0.80 (↑) | $\mathcal{H}$ (↑) |
|---|---|---|---|---|
| | *Reach* | | | |
| PRE | $0.32_{\pm 0.01}$ | $0.31_{\pm 0.00}$ | $0.00/2$ | $0.99_{\pm 0.00}$ |
| RES | $1.00_{\pm 0.00}$ | $1.00_{\pm 0.00}$ | $2.00/2$ | $0.98_{\pm 0.01}$ |
| SP | $0.98_{\pm 0.00}$ | $0.98_{\pm 0.00}$ | $2.00/2$ | $0.97_{\pm 0.00}$ |
| DPPO | $0.93_{\pm 0.01}$ | $0.94_{\pm 0.02}$ | $2.00/2$ | $0.66_{\pm 0.33}$ |
| DPPO[10] | $0.99_{\pm 0.00}$ | $0.99_{\pm 0.00}$ | $2.00/2$ | $0.97_{\pm 0.03}$ |
| | *Lift* | | | |
| PRE | $0.14_{\pm 0.01}$ | $0.15_{\pm 0.01}$ | $0.00/2$ | $0.97_{\pm 0.01}$ |
| RES | $1.00_{\pm 0.00}$ | $0.50_{\pm 0.00}$ | $1.00/2$ | $0.00_{\pm 0.00}$ |
| SP | $0.78_{\pm 0.03}$ | $0.78_{\pm 0.03}$ | $0.67/2$ | $0.98_{\pm 0.01}$ |
| DPPO | $0.99_{\pm 0.01}$ | $0.57_{\pm 0.10}$ | $1.00/2$ | $0.05_{\pm 0.03}$ |
| DPPO[10] | $1.00_{\pm 0.00}$ | $0.56_{\pm 0.08}$ | $1.00/2$ | $0.02_{\pm 0.01}$ |
| | *Avoid* | | | |
| PRE | $0.94_{\pm 0.04}$ | $0.86_{\pm 0.04}$ | $20.00/24$ | $0.63_{\pm 0.00}$ |
| RES | $0.98_{\pm 0.03}$ | $0.04_{\pm 0.00}$ | $1.00/24$ | $0.00_{\pm 0.00}$ |
| SP | $1.00_{\pm 0.01}$ | $0.09_{\pm 0.02}$ | $2.00/24$ | $0.01_{\pm 0.00}$ |
| DPPO | $1.00_{\pm 0.00}$ | $0.26_{\pm 0.11}$ | $6.33/24$ | $0.13_{\pm 0.15}$ |
| DPPO[10] | $1.00_{\pm 0.00}$ | $0.04_{\pm 0.00}$ | $1.00/24$ | $0.00_{\pm 0.00}$ |

**Table 4:** Fine-tuning with regularization (MD-MAD).

| Method | SR (↑) | SR$_M$ (↑) | mc@0.80 (↑) | $\mathcal{H}$ (↑) |
|---|---|---|---|---|
| | *Reach* | | | |
| PRE | $0.32_{\pm 0.01}$ | $0.31_{\pm 0.00}$ | $0.00/2$ | $0.99_{\pm 0.00}$ |
| RES[MD-MAD] | $0.99_{\pm 0.00}$ | $0.99_{\pm 0.00}$ | $2.00/2$ | $1.00_{\pm 0.00}$ |
| SP[MD-MAD] | $1.00_{\pm 0.00}$ | $1.00_{\pm 0.00}$ | $2.00/2$ | $0.97_{\pm 0.01}$ |
| DPPO[MD-MAD] | $0.98_{\pm 0.01}$ | $0.98_{\pm 0.01}$ | $2.00/2$ | $0.67_{\pm 0.43}$ |
| DPPO[10] | - | - | - | - |
| | *Lift* | | | |
| PRE | $0.14_{\pm 0.01}$ | $0.15_{\pm 0.01}$ | $0.00/2$ | $0.97_{\pm 0.01}$ |
| RES[MD-MAD] | $0.99_{\pm 0.00}$ | $0.99_{\pm 0.00}$ | $2.00/2$ | $1.00_{\pm 0.00}$ |
| SP[MD-MAD] | $0.88_{\pm 0.07}$ | $0.88_{\pm 0.07}$ | $1.67/2$ | $0.99_{\pm 0.01}$ |
| DPPO[MD-MAD] | $0.99_{\pm 0.00}$ | $0.55_{\pm 0.07}$ | $1.00/2$ | $0.06_{\pm 0.04}$ |
| DPPO[10] | - | - | - | - |
| | *Avoid* | | | |
| PRE | $0.94_{\pm 0.04}$ | $0.86_{\pm 0.04}$ | $20.00/24$ | $0.63_{\pm 0.00}$ |
| RES[MD-MAD] | $0.99_{\pm 0.01}$ | $0.30_{\pm 0.02}$ | $7.33/24$ | $0.53_{\pm 0.01}$ |
| SP[MD-MAD] | $1.00_{\pm 0.00}$ | $0.42_{\pm 0.00}$ | $10.00/24$ | $0.58_{\pm 0.00}$ |
| DPPO[MD-MAD] | $0.94_{\pm 0.07}$ | $0.43_{\pm 0.05}$ | $9.67/24$ | $0.57_{\pm 0.01}$ |
| DPPO[10] | - | - | - | - |

and (ii) trajectory length asymmetries that bias toward shorter-horizon solutions. Taken together, the results align with the 2D Gaussian mixture experiments and indicate that standard RL fine-tuning progressively destroys multimodality as the reward landscape deviates from the pre-trained trajectory distribution and becomes unbalanced across modes.

**MD-MAD Fine-tuning.** Table 4 shows that incorporating our regularization enables adaptation of the pre-trained policy while largely preserving multimodality, with only minimal trade-offs between diversity and task performance. In *Reach*, regularization leaves success rates unaffected, confirming that our regularization term does not sacrifice performance. In *Lift*, it allows the policy to retain both solution modes from the pre-trained policy, improving over standard fine-tuning. In the more challenging *Avoid*, it sustains high success while preserving a subset of the modes, again outperforming baselines. Although some collapse remains, the results indicate that regularization substantially mitigates mode loss, even under pronounced reward imbalance. Qualitative visualizations of the skills learned by our method are shown in Appendix I.5. Additionally, ablations covering design choices such as curriculum learning and pre-training the steering policy for mode discovery, as well as the role of the regularization weight $\lambda$ and the effect of removing the steering policy after fine-tuning, are reported in Appendix I.1.

## 5.3 ROBOTIC LOCOMOTION AND SEQUENTIAL MANIPULATION

We further evaluate MD-MAD in more challenging settings: ANYmal locomotion, a high-dimensional extension of the 2D Gaussian setup, and the sequential, combinatorial Franka Kitchen task, both detailed in Appendix H. In both cases, we fine-tune the steering policy and compare against the previously introduced baselines, using latent dimensionalities of 4 and 16, respectively.

Table 5 summarizes the locomotion results. Despite the increased dimensionality, the results closely mirror those of the 2D Gaussian Mixture experiments: RES is a strong baseline in this setting, and MD-MAD preserves all four modes of the pre-trained steering policy and prevents collapse during fine-tuning. We further provide visualizations of the fine-tuned policy rollouts and an analysis of the stability of the learned modes in Appendix I.2.

Table 6 reports the results for Franka Kitchen, where success is defined as completing three subtasks in any order. We first observe that the original pre-trained policy exhibits limited mode coverage, as reflected by its low entropy. However, all baselines collapse the multimodality present in the pre-trained policy. In contrast, MD-MAD maintains most of the original behavioral diversity while still fine-tuning the policy to high success. The slight entropy reduction reflects the loss of a single successful sequence, consistent with the sequences reported in Table 14. Exploring different latent

**Table 5:** ANYmal locomotion environment.

| Method | SR ($\uparrow$) | $SR_M$ ($\uparrow$) | mc@0.80 ($\uparrow$) | $\mathcal{H}$ ($\uparrow$) |
|---|---|---|---|---|
| PRE | 0.42 | 0.40 | 0.00/4 | 0.95 |
| RES | 0.96 | 0.97 | 4.00/4 | 0.99 |
| SP | 1.00 | 0.25 | 1.00/4 | 0.00 |
| DPPO[10] | 1.00 | 0.25 | 1.00/4 | 0.00 |
| DPPO | 0.97 | 0.46 | 2.00/4 | 0.27 |
| SP [MD-MAD] | 0.98 | 0.98 | 4.00/4 | 1.00 |

**Table 6:** Franka Kitchen environment.

| Method | SR ($\uparrow$) | $SR_M$ ($\uparrow$) | mc@0.80 ($\uparrow$) | $\mathcal{H}$ ($\uparrow$) |
|---|---|---|---|---|
| PRE | 0.02 | 0.04 | 1.00/24 | 0.44 |
| RES | 1.00 | 0.04 | 1.00/24 | 0.00 |
| SP | 1.00 | 0.04 | 1.00/24 | 0.00 |
| DPPO[10] | 1.00 | 0.04 | 1.00/24 | 0.00 |
| DPPO | 0.71 | 0.04 | 1.00/24 | 0.19 |
| SP [MD-MAD] | 0.98 | 0.08 | 2.00/24 | 0.23 |

dimensionalities could further mitigate this loss and recover the missing mode. Overall, these results demonstrate that MD-MAD effectively preserves multimodal structure in both high-dimensional control tasks and sequential manipulation domains, where the baselines collapse to a single solution.

## 6 CONCLUSIONS, LIMITATIONS AND FUTURE WORK

We studied the problem of fine-tune pre-trained generative policies with RL while preserving multi-modal action distributions. Focusing on diffusion policies trained from demonstrations, we showed that standard fine-tuning often collapsed multimodality to a dominant behavior when the fine-tuning reward landscape diverged from the demonstrations. To address this, we proposed using mutual information as a proxy for multimodality and introduced MD–MAD, an unsupervised mode-discovery method based on a latent reparameterization of a steering policy. We then used the steering policy together with the mutual-information estimate to provide an intrinsic reward that regularized RL fine-tuning toward retaining diverse behaviors. We benchmarked the method across different robotics environments, and showcased that the proposed regularization mitigated mode collapse, supporting MD–MAD as a practical approach to fine-tuning generative policies without sacrificing behavioral diversity.

**Limitations and Future Work.** Our study revealed several trade-offs and open directions. The intrinsic-reward regularization required careful tuning, as excessive weight slowed learning and reduced task success. Maintaining an inference model during fine-tuning also introduced instabilities, as it needed to track the policy's shifting state distribution. Moreover, in its current form, MD-MAD is designed as a task-level mode extractor, as it focuses on discovering and preserving *within-task* behavioral modes, rather than modeling the structure of multi-task datasets. Therefore, scaling MD-MAD to large, heterogeneous datasets may require richer latent parametrizations, such as a hierarchical latent space, to capture multi-task-level multimodality.

Designing the appropriate parametrization and structure of the latent space is also challenging. In all experiments, we employed a single categorical latent $\mathcal{Z}$ indexing discrete behavioral modes within each task. We deliberately used a mildly overparameterized space, and our results indicated that MD-MAD could reliably collapse redundant codes and recover the relevant modes, suggesting that overparameterization is not critical in practice. Nonetheless, exploring more expressive continuous or hybrid latent spaces, together with regularization strategies that improve the controllability of $\mathcal{Z}$, such as entropy or KL terms to balance code usage and mitigate mode collapse, is an important direction for future work.

We occasionally observed that distinct latent codes were mapped to the same environment-defined mode, indicating that our mutual information objective can be sensitive to small variations in the visited state distribution. This is a known weakness of mutual information-based unsupervised skill discovery illustrated in Park et al. (2023), and systematically assessing which trajectory-diversity metrics most effectively retain multimodal behavior is a promising direction for future work.

Finally, although the formulation was independent of language supervision, the learned latent space is amenable to post-hoc semantic grounding. Aligning modes with language via preference learning or VLA mappings and developing a joint inference model that preserves diversity while enabling reliable semantic labels are compelling directions for future work.

# 7 REPRODUCIBILITY STATEMENT

We have made extensive efforts to ensure the reproducibility of our results. All algorithmic details, including model architectures, training procedures, and hyperparameters, are described in the main text and appendix. If any hyperparameter is not explicitly documented in the paper, it will be fully specified in the released code repository. Upon acceptance, we will release the complete codebase, together with configuration files, pretrained checkpoints, and evaluation scripts, to allow exact replication of our experiments. Additionally, proofs of theoretical claims and ablation studies supporting our design choices are included in the appendix.

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

APPENDIX

| Symbol | Description |
|--------|-------------|
| $s_t$ | agent state |
| $a_t^{\mathcal{D}}$ | pre-trained action |
| $a_t^*$ | fine-tuned action |
| $\Delta a_t$ | residual action |

Direct Fine-tuning    Residual Policy    Steering Policy

**Figure 6: Taxonomy of RL Fine-tuning Techniques Discussed in this Work.** Each plot illustrates the learned action-value function $Q(s_t, \cdot)$ as the underlying reward landscape. Direct fine-tuning (left) adapts the pre-trained policy weights to optimize task performance, directly shifting the action distribution toward higher-value regions. Residual policies (center) learn an additive correction $\Delta a_t$ to the pre-trained action $a_t^{\mathcal{D}}$, combining them into a fine-tuned action $a_t^*$. Steering policies (right) learn a policy over the input latent noise of the generative model, biasing sampling toward regions of the noise space whose denoised actions have high-reward behaviors.

## A    EXTENDED RELATED WORK

Robotic manipulation often admits multiple distinct solutions arising from kinematic redundancies, multimodal goals, or heterogeneous demonstrations (Li et al., 2025). We review related work on handling such multimodality in the action distribution from three perspectives: (i) general approaches to learning multimodal behaviors, (ii) fine-tuning of generative policies, where we identify three categories of RL-based techniques schematically illustrated in Figure 6, and (iii) the skill discovery literature, which closely connects to our central idea of unsupervised mode discovery.

### A.1    MULTIMODAL BEHAVIOR LEARNING AND ACTION DIVERSITY

Robotic manipulation often admits multiple distinct solutions, arising from kinematic redundancies, multimodal goals, or heterogeneous demonstrations (Li et al., 2025). Standard RL policies parameterized by unimodal Gaussians collapse to a single behavior, limiting expressiveness and trapping learning in suboptimal modes (Huang et al., 2023). Early work tackled this by introducing a latent-conditioned policy within the policy gradient framework, casting trajectory generation as a latent-variable model to encourage exploration of distinct modes and avoid local minima Huang et al. (2023). Imitation learning and offline RL have built on latent representations to infer discrete behaviors directly from data. Hausman et al. segment unlabeled demonstrations into "intention" clusters and learn a mode-conditioned policy for each cluster (Hausman et al., 2017), while LAPO refines a multimodal policy via an advantage-weighted divergence penalty that preserves original modes during offline finetuning (Chen et al., 2022). Closer to our setting, Habib et al. learn a mixture of low-entropy behavioral tokens by maximizing the discounted cumulative entropy of the mixture policy while minimizing the entropy of each component; however, unlike our method, its focus it to learn mixture components from scratch rather than preserving pre-existing multimodal structure in a pre-trained generative policy. Integrating expressive policy representations such as diffusion and flow-based generative policies further improves upon this by capturing complex, high-dimensional distributions. Deep Diffusion Policy Gradient (DDiffPG) (Li et al., 2024) demonstrated an RL agent that discovers and maintains multiple strategies by parameterizing the policy with a diffusion model. They address the tendency of the greedy RL objective to collapse to one mode by clustering experience and doing mode-specific value learning, thereby ensuring improvement of all discovered modes.

Similar to these approaches, our work builds on a latent-variable model, but we employ it within a steering policy rather than the main policy. Unlike prior approaches that learn multimodal behaviors from scratch, we leverage a pre-trained diffusion model that already encodes diverse demonstrations and focus on fine-tuning it with RL while preserving multimodality in the action distribution.

### A.2    FINE-TUNING OF PRE-TRAINED GENERATIVE POLICIES

Diffusion- and flow-based models provide expressive policy parameterizations for multimodal action distributions, but fine-tuning them with reinforcement learning is challenging due to sequential

sampling and the cost of backpropagating through the generative process. Recent work addresses these issues through three main strategies (illustrated in Figure 6): direct fine-tuning, residual policies, and steering policies. *Direct fine-tuning* approaches adapt the network weights either by distilling the model into a one-step sampler for easier backpropagation (Park et al., 2025; Chen et al., 2024), by casting the denoising process as a sequential decision problem (Ren et al., 2024), or by using differentiable approximations that allow offline Q-learning without backpropagating through all denoising steps (Kang et al., 2023). Despite their promise, such approaches often collapse to a single reward-maximizing mode. *Residual policy* learning methods instead freeze a pre-trained generative policy and learn a small corrective controller via RL to address execution errors (Ankile et al., 2024; Yuan et al., 2024). These techniques, along with careful regularization and architectural choices, can yield substantial performance gains over pure IL, with the potential to preserve the diversity learned from demonstrations. *Steering policy* methods instead bias the sampling process toward high-value actions without modifying the generative model itself. Some methods directly adjust training data or sampled actions using Q-values, either by nudging demonstration actions toward higher values (Yang et al., 2023) or by combining diffusion with Q-learning to bias samples while staying close to the demonstration manifold (Wang et al., 2022). More recently, Wagenmaker et al. (2025) proposed to learn to control the latent noise of generative models, guiding the sampling process toward regions of the noise space whose denoised actions yield higher reward.

Although all these approaches can successfully fine-tune pretrained policies with RL, they lack an explicit mechanism to preserve multimodality, often collapsing to a single reward-maximizing behavior. Our approach extends the steering-policy framework Wagenmaker et al. (2025) by using it not only to bias behavior toward reward but also to uncover and control the latent multimodal structure of a pre-trained diffusion policy. Notably, this perspective positions the steering policy as a complementary module that can be combined with other fine-tuning methods to enforce the retention of diverse behaviors.

### A.3   SKILL DISCOVERY

Multimodal behavior learning has also been explored through the lens of skill discovery methods. The goal of skill discovery is to acquire a set of diverse and distinguishable behaviors without relying on external rewards. A common approach is to maximize mutual information between a latent skill variable and the states or trajectories visited by the policy, as in VIC (Gregor et al., 2016), DIAYN (Eysenbach et al., 2018), VALOR (Achiam et al., 2018), VISR (Hansen et al., 2019), or DADS (Sharma et al., 2019). Other methods rely on successor features (Machado et al., 2017; Hansen et al., 2019), exploration bonuses (Liu & Abbeel, 2021a;b), or hierarchical decompositions (Kim et al., 2021; Zhang et al., 2021) to induce skill diversity.

Most of these works assume training policies from scratch in reward-free settings. However, purely diversity-driven objectives often neglect reward alignment and directed exploration, yielding skills that may not transfer to specific manipulation goals. To mitigate this, previous work has explored a range of approaches such as incorporating language guidance (Rho et al., 2025), combining discovery with generic extrinsic rewards (Emukpere et al., 2024), maximization of hard-to-achieve state transitions (Park et al., 2023), or mutual information maximization between agent and environment sections of state space (Zhao et al., 2021; Cho et al., 2022). Our perspective is different: we leverage a pre-trained model to uncover diverse and useful behaviors already encoded in it. In particular, we are the first to study skill discovery in diffusion policies, where skills are represented as modes in the latent noise space of the generative model.

## B   DERIVATION OF MUTUAL INFORMATION IN LATENT-CONDITIONED POLICIES

We begin by recalling the definition of conditional mutual information between a latent variable $w$ and actions $a$, given states $s$:

$$I(W; A \mid S) := \mathbb{E}_{s \sim p(s)} \left[ \mathbb{E}_{(a,w) \sim p(w,a|s)} \left[ \log \frac{p(w, a \mid s)}{p(a \mid s)\, p(w \mid s)} \right] \right]. \tag{4}$$

In the setting of latent-conditioned policies, we assume a generative process where the state $s \sim p(s)$ is sampled from a fixed distribution, the latent $w \sim p(w)$ is sampled independently of $s$, and actions

are sampled from a conditional policy $\pi(a \mid s, w)$. This induces the joint distribution

$$p(s, w, a) = p(s) \cdot p(w) \cdot \pi(a \mid s, w), \tag{5}$$

and the conditional joint and marginals:

$$p(w, a \mid s) = p(w) \cdot \pi(a \mid s, w), \tag{6}$$
$$p(w \mid s) = p(w). \tag{7}$$

Substituting these expressions into the definition of conditional mutual information, we obtain:

$$
\begin{aligned}
I(W; A \mid S) &= \mathbb{E}_{s \sim p(s)} \Big[ \mathbb{E}_{w \sim p(w),\, a \sim \pi(a|s,w)} \Big[ \log \frac{p(w)\,\pi(a \mid s, w)}{p(w)\,p(a \mid s)} \Big] \Big] \\
&= \mathbb{E}_{s \sim p(s)} \Big[ \mathbb{E}_{w \sim p(w),\, a \sim \pi(a|s,w)} \Big[ \log \frac{\pi(a \mid s, w)}{p(a \mid s)} \Big] \Big].
\end{aligned}
\tag{8}
$$

Recognizing this expression as the Kullback–Leibler (KL) divergence between the conditional distribution $\pi(a \mid s, w)$ and its marginal $p(a \mid s)$, we rewrite the mutual information as:

$$I(W; A \mid S) = \mathbb{E}_{s \sim p(s)} \big[ \mathbb{E}_{w \sim p(w)} \big[ D_{\mathrm{KL}}\big( \pi(a \mid s, w) \,\|\, p(a \mid s) \big) \big] \big]. \tag{9}$$

In this formulation, $p(a \mid s)$ is interpreted as the marginal action distribution under latent sampling:

$$p(a \mid s) = \mathbb{E}_{w \sim p(w)} \big[ \pi(a \mid s, w) \big]. \tag{10}$$

This derivation provides a formal and tractable characterization of the mutual information between latent variables and actions under a latent-conditioned policy. It also justifies the use of mutual information as a measure of multimodality: if $w$ has a significant influence on the action distribution $\pi(a \mid s, w)$, then the divergence between conditionals and the marginal $p(a \mid s)$ is large, leading to a high $I(W; A \mid S)$. Conversely, if the latent has little effect on the action distribution, the mutual information approaches zero.

## C    Multimodality Implies Positive Mutual Information

**Proposition 1.** *Let $W, A$ and $S$ be discrete random variables such that*

$$P_{A|S,W}(\cdot | s_0, w_1) \neq P_{A|S,W}(\cdot | s_0, w_2)$$

*for some $s_0, w_1, w_2$ with $P_S(s_0) P_W(w_1) P_W(w_2) > 0$. Then $I(W; A|S) > 0$.*

*Proof.* First we show that for any discrete probability distributions $P$ and $Q$ on a common sample space $\mathcal{X}$,

$$P \neq Q \qquad \Rightarrow \qquad D_{\mathrm{KL}}(P \,\|\, Q) > 0. \tag{11}$$

Indeed, as $\log$ is concave,

$$-D_{\mathrm{KL}}(P \,\|\, Q) = \sum_{x:P(x)>0} P(x) \log \frac{Q(x)}{P(x)} \overset{\text{(a)}}{\leq} \log \left( \sum_{x:P(x)>0} P(x) \frac{Q(x)}{P(x)} \right) \overset{\text{(b)}}{\leq} 0.$$

Moreover, either:

- $Q(x_1)/P(x_1) \neq Q(x_2)/P(x_2)$ for some $x_1, x_2$ in the support of $P$, and as $\log$ is strictly concave it follows that inequality (a) is strict;

- or $Q(x)/P(x)$ is constant for $x$ in the support of $P$, and as $P \neq Q$ it follows that $\sum_{x:P(x)>0} Q(x) < 1$ so inequality (b) is strict.

Therefore (11) holds.

Let $i^* \in \{1, 2\}$ be an index such that $P_{A|S}(\cdot|s_0) \neq P_{A|S,W}(\cdot|s_0, w_{i^*})$, noting that such an $i^*$ exists by the hypothesis that the distributions $P_{A|S,W}(\cdot|s_0, w_i)$ for $i = 1, 2$ are distinct. Using the expression for $I(W; A|S)$ of Appendix [YourRef], and the nonnegativity of KL-divergence,

$$I(W; A|S) = \mathbb{E}_S \mathbb{E}_W [D_{\mathrm{KL}}(P_{A|S,W}(\cdot|S, W) \| P_{A|S}(\cdot|S))]$$
$$\geq P_S(s_0) P_W(w_{i^*}) D_{\mathrm{KL}}(P_{A|s_0, w_{i^*}}(\cdot|s_0, w_{i^*}) \| P_{A|S}(\cdot|s_0))$$
$$> 0$$

where in the last line we used the hypothesis that $P_S(s_0) P_W(w_1) P_W(w_2) > 0$ and equation (11). $\qquad\square$

## D  METHOD DETAILS

### D.1  CONNECTION TO SKILL DISCOVERY.

The variational lower bound in equation 2 is formally analogous to those used in prior skill discovery methods, but its purpose in our setting is fundamentally different. In mutual-information-based skill discovery, the bound is optimized jointly with the policy to encourage exploration and broaden coverage of the state space. By contrast, our diffusion policy is pre-trained and fixed, so the mutual information objective cannot alter the state distribution. Instead, maximizing $I(Z; S)$ serves to uncover and control the intrinsic multimodality already embedded in the generative policy by promoting diversity in the action space $\mathcal{A}$. In addition, this bound provides a practical metric to quantify multimodality in pre-trained policies, as demonstrated in Section 5.1.

### D.2  CURRICULUM LEARNING.

Unlike in standard skill discovery, we have access to full trajectory rollouts for each mode we want to discover. However, this makes the joint optimization of the steering policy and inference model challenging as the policy must maintain temporal consistency while producing behaviors that remain discriminable by the inference model $q_\phi$, which can lead to instability during training. To mitigate this, we introduce a curriculum strategy that gradually increases the trajectory horizon. Concretely, instead of unrolling episodes for the full environment length $T$ from the outset, we begin training with shorter horizons $H < T$ and progressively extend them until reaching the maximum length. This staged schedule eases the optimization by allowing the policy to first acquire locally consistent behaviors, before being required to sustain them over longer time horizons, thereby improving the stability and quality of the learned latent modes. The proposed curriculum is visualized in Figure 7.

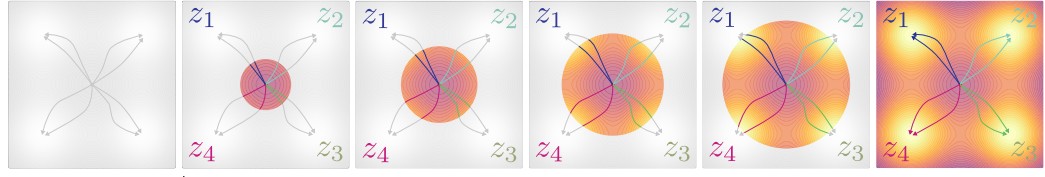

**Figure 7: Curriculum Learning.** Illustration of the curriculum strategy in a toy environment with four discrete modes. The environment is defined by a mixture of four Gaussian modes (details in Section 5.1), each corresponding to a distinct cluster of trajectories. Starting from short horizons, the inference model $q_\phi$ only needs to discriminate local trajectory prefixes, which simplifies learning. As the horizon gradually increases, the trajectory distributions expand, and the modes become more separable across the state-action space. The curriculum thus enables the steering policy to develop temporally consistent and discriminable behaviors, progressively uncovering the underlying latent structure of the pre-trained model.

### D.3  ALGORITHM

We outline here Algorithm 1. We begin from the pre-trained diffusion policy $\pi_\theta(a \mid s, w)$ and initialize the steering policy $\pi_\psi^{\mathcal{W}}(w \mid s, z)$, inference model $q_\phi(z \mid s, a)$, and critic $V_\omega(s, z)$, with

intrinsic scale $\lambda \geq 0$, uniform prior $p(z)$, epochs $E$, episodes per epoch $N$, warm-start $E_{\text{wp}}$, initial horizon $H_0$, max horizon $T$, and a scheduler $H(e) \in [H_0, T]$ that increases the rollout horizon by a fixed step every 20 epochs after a first warm-up of 100 epochs. For each epoch $e$ and episode $n$, we sample a latent $z \sim p(z)$ once and keep it fixed over the rollout of length $H(e)$; at each step we draw $w_t \sim \pi_\psi^{\mathcal{W}}(w \mid s_t, z)$, then $a_t \sim \pi_\theta(a \mid s_t, w_t)$, and transition $s_{t+1} \sim p(\cdot \mid s_t, a_t)$. The intrinsic reward is $r_t^{\text{int}} = \lambda\big(\log q_\phi(z \mid s_{t+1}, a_t) - \log p(z)\big)$. During the *mode-discovery* stage ($e < E_{\text{wp}}$) we optimize using intrinsic-only returns $r_t^{\text{tot}} = r_t^{\text{int}}$, allowing the curriculum $H(e)$ to grow from $H_0$ toward $T$ so the policy first attains locally consistent behaviors before sustaining them over longer horizons. After the warm-start ($e \geq E_{\text{wp}}$), we introduce the task reward and train with $r_t^{\text{tot}} = r_{\text{env}}(s_t, a_t) + r_t^{\text{int}}$ to steer toward high-return regions without collapsing diversity. At the end of each epoch, we update the actor and critic with PPO, minimizing $L_\pi^{\text{PPO}}(\psi) + c_V L_V(\omega) + c_{\mathcal{H}} L_{\mathcal{H}}(\psi)$, and train the inference model by NLL, $\min_\phi L_q(\phi) = -\mathbb{E}[\log q_\phi(z \mid s, a)]$; this repeats for $e = 1, \ldots, E$ with horizon scheduling and the stage switch as specified.

# E  IMPLEMENTATION DETAILS

We now detail the implementation and training of the pre-trained policy, all the baseline policies, and the discriminator. We also describe how our method integrates with these general fine-tuning strategies. All approaches employ PPO as fine-tuning RL algorithm with clipping parameter $\epsilon = 0.2$, GAE $\lambda = 0.95$, discount $\gamma = 0.99$, and Adam with learning rate $3 \times 10^{-4}$. To facilitate reproducibility, we will release the full codebase together with all hyperparameters required to reproduce the results reported in this paper.

## E.1  PRE-TRAINED POLICY AND DPPO FINE-TUNING

The diffusion policy is trained with the standard behavioral cloning objective for diffusion models, where the network predicts the injected noise conditioned on the noisy actions. We follow the implementation and hyperparameter setup of DPPO Ren et al. (2024), using a cosine noise schedule during training. The action horizon coincides with the execution horizon and consists of 4 action steps per chunk. Pre-training is performed with 20 denoising steps, while inference uses DDIM (Song et al., 2020) sampling with 2 steps. For frozen policies, we set $\eta = 0$, whereas for fine-tuning, we set $\eta = 1$, which is equivalent to applying DDPM (Ho et al., 2020). This choice ensures steerability of the policy and avoids memoryless noise schedules. The policy head is implemented as a multi-layer perceptron (MLP) with hidden dimensions $\{512, 512, 512\}$, and a time-embedding dimension of 16, which we found to improve training stability compared to UNet backbones, similar to Ren et al. (2024). For fine-tuning, we follow the implementation and hyperparameters introduced in Ren et al. (2024), with the only addition of decreasing the number of fine-tuning steps of the denoising process form 10 to 2 to ensure non-memoryless noise schedule.

## E.2  RESIDUAL POLICY

The residual policy learns an additive correction to the action chunk $a_{t:t+H}$ of length $H$ proposed by the pre-trained diffusion policy, such that $a_{t:t+H}^* = a_{t:t+H} + \lambda \Delta a_{t:t+H}$. Concretely, the residual network receives as input the state and the pre-trained action chunk, and outputs a correction term that is passed through a $\tanh$ activation to ensure bounded updates, $\pi^{\text{RES}}(\Delta a_{t:t+H} \mid s_t, a_{t:t+H})$. To prevent the residual from completely overriding the original action, its contribution is scaled by a tunable factor $\lambda$, which balances task success with fidelity to the pre-trained behavior. This scaling parameter is selected following prior work and tuned empirically to trade off between preserving the original action distribution and improving task success rates. The residual policy is implemented as a Gaussian policy parameterized by a multilayer perceptron with hidden layers of dimension $\{256, 256, 256\}$ and Mish activations.

## E.3  STEERING POLICY

The steering policy $\pi_\psi^{\mathcal{W}}(w \mid s, z)$ is implemented as a Gaussian policy parameterized by an MLP with hidden layers of size $\{256, 256, 256\}$. To constrain its support within that of the original diffu-

sion prior, we apply a KL regularization during training of the form

$$\mathcal{L}_{\mathrm{KL}} = \mathbb{E}_{s,z}\Big[ D_{\mathrm{KL}}\big(\pi_\psi^{\mathcal{W}}(w \mid s, z) \,\big\|\, \mathcal{N}(0, I)\big) \Big],$$

where $\mathcal{N}(0, I)$ denotes the isotropic Gaussian prior used in the diffusion model. The latent variable $z \in 0, 1, \ldots, K-1$ is sampled from a uniform categorical prior $p(z)$, as we empirically found discrete latents easier to learn and more stable than continuous ones. The dimensionality of the latent space is a hyperparameter, in the experiments we consider $K = \{4, 8, 16\}$. Training proceeds in two stages: for the first 200 epochs, the steering policy is optimized only with the intrinsic reward $\log q_\phi(z \mid s, a) - \log p(z)$, serving as a mode-discovery phase; in the remaining epochs, the environment reward is added to steer behaviors toward high-return regions while retaining multimodality.

### E.4 INFERENCE MODEL

The inference model $q_\phi(z \mid s)$ is implemented as a categorical classifier over the latent codes $z \in \{0, \ldots, K-1\}$. It consists of a multilayer perceptron with hidden layers of dimension $\{256, 256, 256\}$, Mish activations (Misra, 2019), and a final softmax output producing the class probabilities $q_\phi(z \mid s)$. To prevent overfitting to small variations in continuous states, Gaussian noise with standard deviation $\{1.0, 0.01, 0.001\}$ (depending on the task) is injected into the inputs during training only. The model is trained by minimizing the negative log-likelihood $\mathcal{L}_{\mathrm{NLL}}(\phi) = -\mathbb{E}_{(s,a,z)}\big[\log q_\phi(z \mid s)\big]$, where the expectation is taken over state-action pairs generated by the steering policy and latent codes sampled from the prior $p(z)$. During training of the steering policy, the log-posterior $\log q_\phi(z \mid s)$ serves as an intrinsic reward, combined with the prior correction term $-\log p(z)$, thereby providing the intrinsic objective for mode discovery and diversity-preserving fine-tuning.

### E.5 INTEGRATING WITH OTHER FINE-TUNING TECHNIQUES.

The steering policy with mode discovery uncovers and controls the behavioral modes of the pre-trained diffusion mode, steering them toward regions of high reward. However, because this mechanism does not update the diffusion weights directly, its performance remains bounded by the expressiveness of the pre-trained policy. From this perspective, the steering policy can be viewed as an *exploration agent* that guides state visitation in a structured way, and can therefore be seamlessly combined with existing fine-tuning methods discussed in Section 2. A key distinction is that our framework provides access to a discriminator that evaluates whether the fine-tuned behaviors remain consistent with the discovered modes, supplying an intrinsic reward that discourages collapse into a single strategy. While the steering policy itself can continue to adapt jointly with the diffusion model, we found it beneficial to update the discriminator with a very low learning rate: this allows it to accommodate novel states encountered during fine-tuning while preserving the previously identified mode structure, thereby stabilizing multimodality retention.

## F  BASELINE METHODS AND EVALUATION METRICS DISCUSSION

Following the characterization introduced in Section A.2, we benchmark our approach against representative strategies for on-policy fine-tuning of generative policies, focusing on diffusion models but noting that analogous evaluations apply to flow-matching policies. Specifically, we consider methods that do (i) direct fine-tuning, (ii) residual corrections, and (iii) steering, noting that none of these explicitly seek to preserve multimodality. As a direct fine-tuning approach, we include `DPPO` (Ren et al., 2024), which optimizes the diffusion policy weights with PPO. We consider the DDIM parameterization of the generative process to ensure non-memoryless noise schedules, while maintaining a balance between $\eta > 0$ and the number of reverse diffusion steps to facilitate weight fine-tuning. To examine the effect of decreasing the number of reverse diffusion steps, we also consider the original hyperparameters of the `DPPO` baseline that uses the full denoising chain for action sampling with DDPM parameterization, and fine-tunes the last 10 steps, denoted `DPPO[10]`, which makes the generation process non-memoryless.

As a residual fine-tuning approach (`RES`), we evaluate Policy Decorator (Yuan et al., 2024), where a lightweight residual network is trained on top of the frozen pre-trained diffusion model. This allows task adaptation while limiting catastrophic interference with the base model. Finally, we

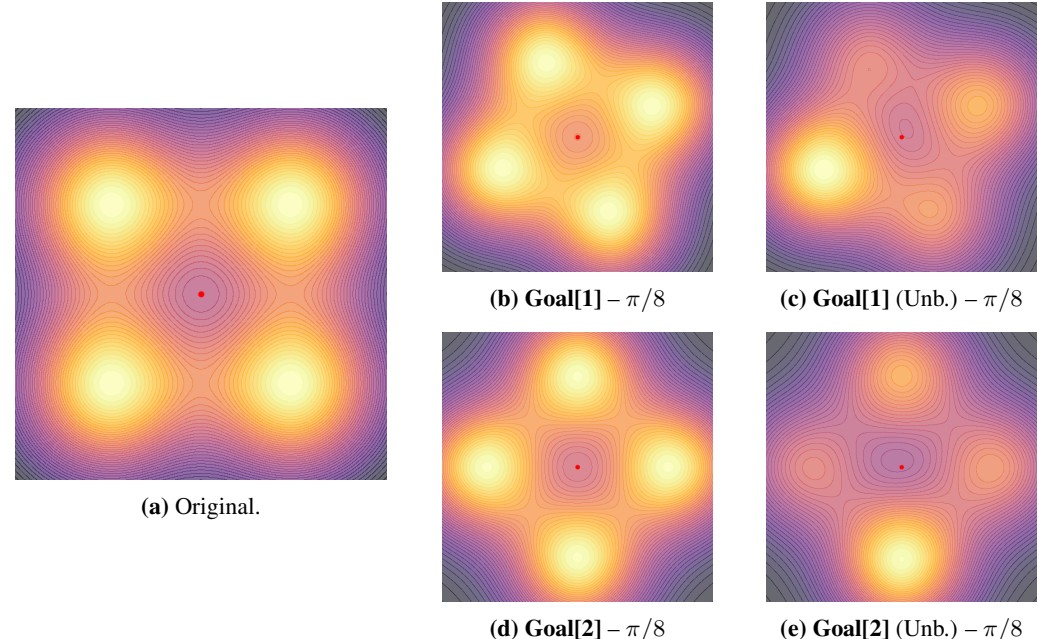

**Figure 8:** Reward landscapes: (a) Original environment; (b–e) rotated goal variants with balanced and unbalanced setups.

consider Wagenmaker et al. (2025) as a steering-based policy `SP`, which adapts the latent noise distribution $w$ to bias the pre-trained policy toward high-reward behaviors. This category operates entirely in the latent space and, like the others, does not include any explicit mechanism for mode discovery or diversity preservation.

Importantly, our approach is orthogonal to these categories: the proposed multimodality-preserving regularizer can be combined with either residual or steering-based fine-tuning under non-memoryless noise schedules. Accordingly, we report results both for the standalone baselines and for their variants augmented with our multimodality regularizer, denoted as `X[MD-MAD]`, where `X` indicates the corresponding baseline. Full implementation details for all baselines and their regularized variants are provided in Appendix E.

**Evaluation Metrics.** We assume access to the ground truth modes of the trajectories executed by the policy in simulation. and we evaluate fine-tuned policies along two axes: *task success* and *behavioral diversity*. For task success, we report the overall success rate SR, and two mode-aggregated success measures: the success rate weighted for each mode $\text{SR}_{\text{M}} = \frac{1}{K}\sum_{i=1}^{K} \text{SR}_i$, which guards against degenerate solutions (e.g. $100\%$ success on a single mode but failure on others), and mode coverage $\text{mc@}\tau = \frac{1}{K}\sum_{i=1}^{K} \mathbf{1}\{\text{SR}_i \geq \tau\}$, the fraction of modes solved above threshold $\tau$.

To further measure multimodality, we follow the D3IL benchmark (Jia et al., 2024) and compute the entropy of the empirical distribution over modes among all rollouts: $H(\pi) = -\sum_{i=1}^{K} p_i \log p_i$, where $p_i$ is the fraction of episodes in mode $i$. A higher entropy reflects more balanced usage of the available modes, whereas a reduction after fine-tuning is indicative of mode collapse. All metrics are computed from $N = 1024$ evaluation episodes with fixed seeds for fair comparison, and we report both the mean and standard deviation over three independent runs with different random seeds.

# G   2D GAUSSIAN MIXTURE ENVIRONMENT

We provide in this section detailed information regarding the implementation of the 2D Gaussian mixture environment, as well as ablation evaluation on the dimensionality of the latent space, the structure learned by the steering policy, and the effect of removing the steering policy after fine-tuning.

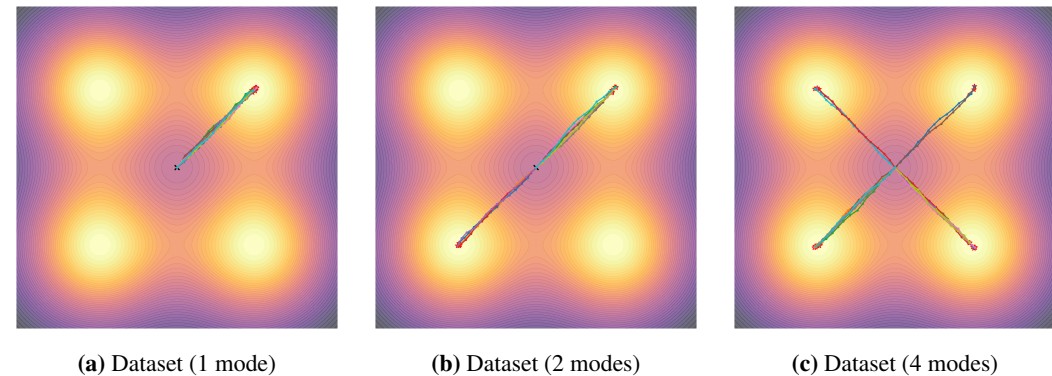

**(a)** Dataset (1 mode)          **(b)** Dataset (2 modes)          **(c)** Dataset (4 modes)

**Figure 10:** Expert datasets with different multimodal behaviors used to pre-train diffusion models to investigate mutual information as a proxy of multimodality.

### G.1 IMPLEMENTATION DETAILS

We designed a two-dimensional navigation task where the reward landscape is given by a mixture of 4 Gaussians. The agent's state is its position $(x, y) \in \mathbb{R}^2$, initialized at the origin $(0, 0)$. Actions are modeled as displacements $(\Delta x, \Delta y)$ applied at each step. The instantaneous reward at position $\text{pos} = (x, y)$ is defined as

$$r(x, y) = \sum_{(c_x, c_y) \in \mathcal{C}} \exp\left(-\frac{(x - c_x)^2 + (y - c_y)^2}{2\sigma^2}\right), \tag{12}$$

where $\mathcal{C}$ is the set of goal centers and $\sigma$ controls the spread of each Gaussian mode. An episode is successful if the agent reaches within a fixed distance of any goal center.

We consider two variants of this reward landscape:

- **Balanced landscape.** Each Gaussian mode contributes equally to the reward. This creates a symmetric multimodal environment where all goal regions are equally attractive.

- **Unbalanced landscape.** To introduce variability in mode prominence, we assign each Gaussian a random weight $w_i \sim \mathcal{U}(0, 1)$. To avoid degenerate scaling while preserving relative preferences, the weights are normalized via a softmax transformation, i.e.

$$\tilde{w}_i = \frac{\exp(w_i)}{\sum_j \exp(w_j)},$$

  and the reward is defined as $r(x, y) = \sum_i \tilde{w}_i \exp\left(-\frac{(x - c_x^{(i)})^2 + (y - c_y^{(i)})^2}{2\sigma^2}\right)$. This ensures that all modes remain present but with uneven reward magnitudes, yielding a more challenging and realistic multimodal landscape.

We refer to these as the **unbalanced Goal[1]** and **unbalanced Goal[2]** environments. Figure 8 provides visualizations of all balanced and unbalanced variants.

### G.2 EXPERT DEMONSTRATIONS

Figure 10 shows the expert demonstration dataset used for the experiments in section 5.1.

### G.3 DIMENSIONALITY OF $\mathcal{Z}$

We next examine the effect of the latent dimensionality $|\mathcal{Z}|$ on multimodality preservation. We repeat the **Goal[2]** evaluation using the RES and DPPO baselines with mode discovery, varying the number of latent codes.

Results are reported in Table 7. A dimension of $|\mathcal{Z}| = 4$, which matches the ground-truth number of modes, fails to fully capture all task modalities. This limitation stems from our inference model, which distinguishes modes through state coverage and can become sensitive to minor state variations, occasionally treating nearby but distinct states as different modes. Increasing dimensionality ($|\mathcal{Z}| = 8, 16$) improves coverage by promoting exploration of diverse trajectories. However, excessively large latent spaces introduce in-

**Table 7: Ablation on the dimensionality of $\mathcal{Z}$.**

| | Goal [2] | | | |
|---|---|---|---|---|
| **Method** | SR | $SR_M$ | mc@0.80 | $\mathcal{H}$ |
| | $|\mathcal{Z}| = 4$ | | | |
| RES[MD-MAD] | $1.00 \pm 0.00$ | $0.75 \pm 0.00$ | $3.00/4$ | $0.74 \pm 0.00$ |
| DPPO[MD-MAD] | $1.00 \pm 0.00$ | $0.75 \pm 0.00$ | $3.00/4$ | $0.74 \pm 0.00$ |
| | $|\mathcal{Z}| = 8$ | | | |
| RES[MD-MAD] | $1.00 \pm 0.00$ | $1.00 \pm 0.00$ | $4.00/4$ | $0.92 \pm 0.00$ |
| DPPO[MD-MAD] | $0.64 \pm 0.45$ | $0.63 \pm 0.45$ | $2.33/4$ | $0.99 \pm 0.00$ |
| | $|\mathcal{Z}| = 16$ | | | |
| RES[MD-MAD] | $1.00 \pm 0.00$ | $1.00 \pm 0.00$ | $4.00/4$ | $0.94 \pm 0.00$ |
| DPPO[MD-MAD] | $0.79 \pm 0.00$ | $0.82 \pm 0.00$ | $2.00/4$ | $0.94 \pm 0.00$ |

efficiencies: for instance, DPPO[MD-MAD] deteriorates at $|\mathcal{Z}| = 16$, likely due to a trade-off between task optimization and diversity. These results suggest that latent dimensionality should be tuned to the complexity of the multimodal structure, and that more robust inference models beyond simple state coverage may further improve mode discovery, representing an interesting direction for future work.

### G.4  STRUCTURE INDUCED IN THE LATENT SPACE

We investigate what the structure learned by the steering policy is in the policy latent space. We probe what the steering policy actually learns by inspecting the input-noise latents it predicts, rather than the trajectories executed by the full policy. Concretely, for the initial state $s_0$ and each skill label $z \in \{0, 1, 2, 3\}$, we draw 1024 samples $w \sim \pi_\psi^{\mathcal{W}}(w \mid s_0, z)$ and visualize them in Figure 11 together with kernel-density contours and the per-skill mean. The figure reveals a clear four-cluster organization where each skill forms a compact, well-separated mode in the latent space, with only limited cross-skill overlap. This analysis shows that the steering head has learned a discrete, multimodal latent structure aligned with the modes present in the original demonstration dataset.

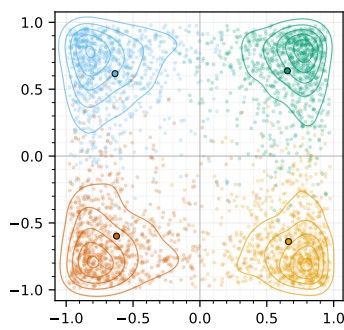

**Figure 11:** Latent noise samples $w$ for $z \in \{0, 1, 2, 3\}$.

## H  TASKS DESCRIPTION

We evaluate our approach on five robotics tasks. These include three robotic manipulation tasks and one locomotion tasks implemented within the ManiSkill (Tao et al., 2024) framework: *Reach*, *Lift*, *Avoid* (re-implemented from D3IL (Jia et al., 2024)), as manipulation tasks and *ANYmal* as locomotion task. We further integrate the *Franka Kitchen* environment from D4RL (Fu et al., 2020) as a sequential multi-task setting. Each task (shown in Figure 12) exhibits distinct forms and degrees of multimodality. In all cases, multimodality refers to the existence of several equally valid but spatially different solution strategies that the robot may follow to achieve the task. Multimodality arises either from goal diversity or, for a fixed goal, from multiple feasible trajectories that lead to successful completion. All manipulation tasks are performed with a Franka Emika Panda robot, where agent actions are parameterized as 6-DoF end-effector delta poses ($\Delta x, \Delta y, \Delta z, \Delta roll, \Delta pitch, \Delta yaw$). The action space for the locomotion task consists of the 12 delta joint position commands controlling the four legs of the ANYmal.

**Reach (2 modes).**  In *Reach*, the agent must contact a green sphere while avoiding a gray obstacle; success can be achieved by approaching from either side. These left–right approaches constitute two disjoint solution classes, creating a simple bimodal structure. This task is comparatively simple, as multimodality appears only at the beginning of the trajectory, after which the policy is effectively committed to a single mode. The state space comprises the robot joint positions and velocities, the

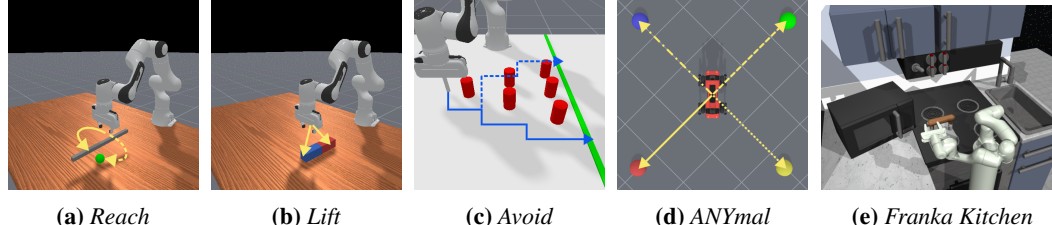

**(a)** *Reach*  **(b)** *Lift*  **(c)** *Avoid*  **(d)** *ANYmal*  **(e)** *Franka Kitchen*

**Figure 12:** Visualization of the five ManiSkill tasks used in our evaluation. For each task, except the *Franka Kitchen*, we highlight representative modes for solving the task.

end-effector pose, as well as goal and bar poses. The maximum episode length is 100 steps. The task is considered to be successful if the agent reaches the goal within a pre-defined threshold

**Lift (2 modes).** In *Lift*, the agent must lift a peg into vertical position. The peg can be grasped and lifted upright from either the red or blue side, yielding multiple valid grasping strategies. Here, multimodality reflects the existence of several grasp affordances around the object, which lead to distinct grasp–lift trajectories even when the final goal is identical. The initial randomization of object configurations increases the ambiguity and difficulty of separating modes. The state space comprises the robot joint positions and velocities, the end-effector pose, as well as the peg pose. The maximum episode length is 200 steps. The task is considered to be successful if the peg is successfully lifted (assessed through the pose of the object) and stable.

**Avoid (24 modes).** In the *Avoid* task, the agent must cross the green line by avoiding the obstacles in the table. This is the most challenging as numerous modalities emerge later in the trajectory, each corresponding to a distinct avoidance strategy with different path lengths. In this case, only the initial end-effector position is randomized at reset, while the obstacle remains fixed, emphasizing the diversity of possible avoidance strategies. The state representation encompasses the end-effector's desired position and actual position in Cartesian space, with the caveat that the robot's height (z position) remains fixed. The actions are represented by the desired velocity of the robot along the x and y axis. The maximum episode length is 300 steps. The task is considered to be successful if the robot-end-effector reaches the green finish line.

**ANYmal Locomotion (4 modes).** In this locomotion task, a quadrupedal ANYmal robot must navigate to one of four goal locations placed at fixed positions in the environment. Multimodality arises from goal diversity: each goal corresponds to a distinct target direction and therefore induces different optimal locomotion strategies and turning behaviors. Demonstrations are generated by training four separate RL agents, each optimized to reach a single goal, producing unimodal expert trajectories for each target. When combined, these demonstrations form a four-modal behavior distribution that the policy must preserve. The state space includes proprioceptive robot observations (joint states, base pose, and velocities) and the relative goal position with respect to the agent position. The maximum episode length is 200 steps, and an episode is successful if the robot reaches any goal within a predefined tolerance.

**Franka Kitchen (24 modes).** The Franka Kitchen environment from D4RL (Fu et al., 2020) contains demonstrations of a robot manipulating several articulated objects (microwave, kettle, burner, light switch). We train on the mixed demonstration dataset, which contains trajectories performing different task combinations in varying orders, but never completing all four evaluation subtasks sequentially. As a result, multimodality emerges both from the diversity of partial task orders and from multiple valid ways to interact with each object. For evaluation, we follow the common benchmark and consider four subtasks—`microwave`, `kettle`, `bottom burner`, `light switch`. Success is achieved when the policy completes three of the four subtasks within the episode, possibly in any order. The state space consists of robot joint states, end-effector pose, and object poses; the maximum horizon is 280 steps.

All environments provide dense or intermediate reward functions to support fine-tuning, and we employ a heuristic to identify the mode associated with each trajectory, enabling consistent evalu-

**Table 8:** Ablation experiments on design choices.

| Method | SR | $SR_M$ | mc@0.80 | $\mathcal{H}$ |
|---|---|---|---|---|
| PRE | $0.14_{\pm 0.01}$ | $0.15_{\pm 0.01}$ | $0.00/2$ | $0.97_{\pm 0.01}$ |
| RES[MD-MAD] | $0.99_{\pm 0.00}$ | $0.99_{\pm 0.00}$ | $2.00/2$ | $1.00_{\pm 0.00}$ |
| RES[NO-PRE MD-MAD] | $0.91_{\pm 0.04}$ | $0.79_{\pm 0.11}$ | $1.33/2$ | $0.74_{\pm 0.08}$ |
| RES[NO-FT MD-MAD] | $0.00_{\pm 0.00}$ | $0.00_{\pm 0.00}$ | $0.00/2$ | $0.00_{\pm 0.00}$ |
| RES[NO-CURR MD-MAD] | $0.85_{\pm 0.08}$ | $0.83_{\pm 0.08}$ | $1.33/2$ | $0.95_{\pm 0.05}$ |

**Table 9:** Ablation experiment on removing the steering policy after fine-tuning with MD-MAD

| Method | SR | $SR_M$ | mc@0.80 | $\mathcal{H}$ |
|---|---|---|---|---|
| PRE | $0.14_{\pm 0.01}$ | $0.15_{\pm 0.01}$ | $0.00/2$ | $0.97_{\pm 0.01}$ |
| *With Steering Policy* | | | | |
| RES[MD-MAD] | $0.99_{\pm 0.00}$ | $0.99_{\pm 0.00}$ | $2.00/2$ | $1.00_{\pm 0.00}$ |
| DPPO[MD-MAD] | $0.99_{\pm 0.00}$ | $0.55_{\pm 0.07}$ | $1.00/2$ | $0.06_{\pm 0.04}$ |
| *Without Steering Policy (Random Sampling)* | | | | |
| RES[MD-MAD] | $0.95_{\pm 0.02}$ | $0.94_{\pm 0.02}$ | $2.00/2$ | $0.93_{\pm 0.03}$ |
| DPPO[MD-MAD] | $0.99_{\pm 0.00}$ | $0.58_{\pm 0.06}$ | $1.00/2$ | $0.08_{\pm 0.03}$ |

ation of multimodality. Additional implementation details will be available upon the release of the codebase.

# I  ABLATION EXPERIMENTS

## I.1  METHOD ABLATIONS

We first study the effect of the regularization weight $\lambda$ on task performance, focusing on the *Lift* task with the RES[MD-MAD] baseline. Figure 13 shows that as $\lambda$ increases, the intrinsic reward increasingly dominates over the task reward, leading to a drop in success rate. This illustrates the trade-off: stronger regularization favors diversity at the expense of task performance.

Next, we analyze the impact of (i) pre-training with only the mode-discovery reward ([NO-FT MD-MAD]) and (ii) omitting fine-tuning of the inference model and steering policy when adapting the main policy with another fine-tuning technique ([NO-PRE MD-MAD]), (iii)

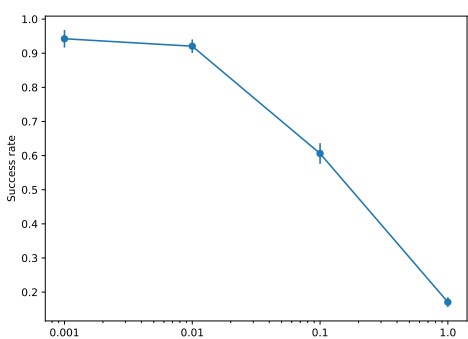

**Figure 13:** Impact of the regularization coefficient $\lambda$ on the task success rate.

removing the curriculum stage during the mode-discovery phase ([NO-CURR MD-MAD]). These ablations, reported in Table 8) for the *Lift* task with RES[MD-MAD], reveal that all factors negatively affect performance. In particular, disabling fine-tuning of the inference model and steering policy is catastrophic: the mutual-information signal becomes uninformative as the policy is driven toward out-of-distribution states relative to pre-training.

Finally, we evaluate whether policies fine-tuned with MD-MAD retain multimodality and performance once the steering head is removed, i.e., actions are again driven by the original latent

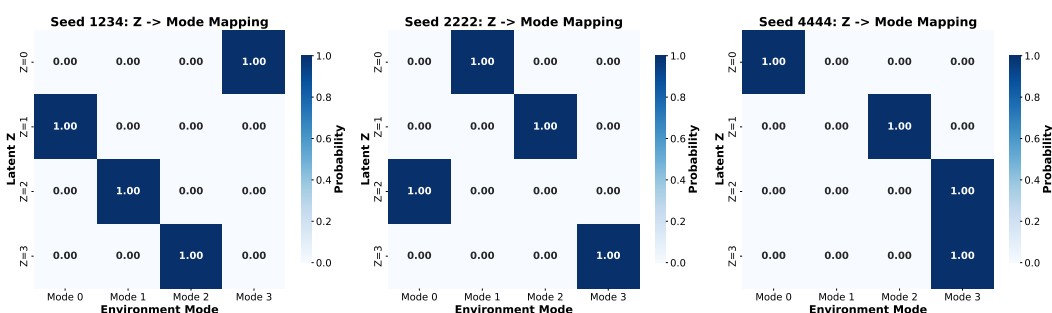

**Figure 14:** Confusion matrices of the mappings from the latent $z \in \mathcal{Z}$ to the ground truth environment's modes.

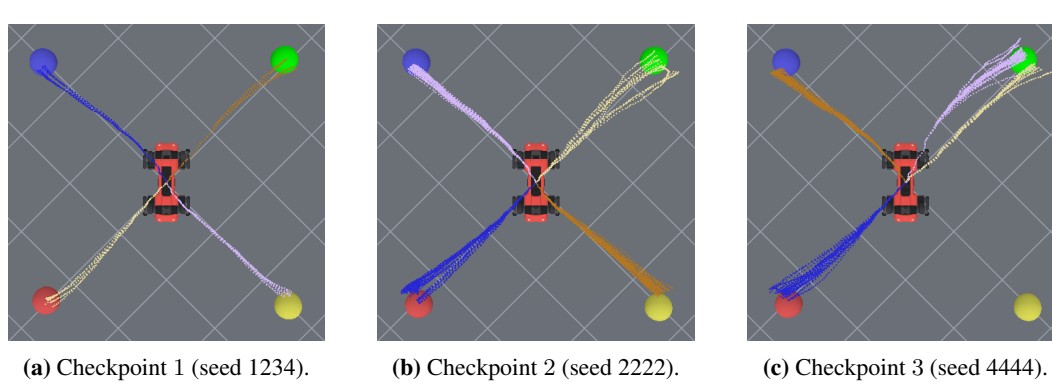

(a) Checkpoint 1 (seed 1234).    (b) Checkpoint 2 (seed 2222).    (c) Checkpoint 3 (seed 4444).

**Figure 15:** Qualitative visualization of the trajectories distributions for different checkpoints, where different colors correspond to different $z \in \mathcal{Z}$.

noise prior. Table 9 reports success and multimodality metrics for only the `DPPO[MD-MAD]` and `RES[MD-MAD]` on *Lift*, as removing the steering on the `SP` baseline would regress the performance back to the original pre-trained policy. The residual baseline shows minimal degradation after removing the steering head, indicating that residual updates internalize the discovered modes into the policy. Similarly, `DPPO[MD-MAD]` exhibits similar performance with respect to the version including the steering head.

We hypothesize that MD-MAD's regularization on the steering output, penalizing deviations from the original normal noise, encourages compatibility between the learned behaviors and the base diffusion noise. During fine-tuning, steering guides exploration over $z$ to expose distinct modes, while the regularizer keeps the induced noise close to the prior, allowing the policy to absorb mode structure without depending on explicit steering at inference. Consequently, `RES[MD-MAD]` especially, can execute diverse behaviors when sampling from the unmodified prior, preserving multimodality with limited impact on task success and making it a strong candidate for fine-tuning generative policies.

### I.2   MODE STABILITY ANALYSIS

To assess the stability of the latent steering variable discovered by MD-MAD, we evaluate whether different training seeds induce consistent mappings between latent codes and environment modes in the *ANYmal* task. We consider 3 different seeds, all steering policies were trained under identical conditions and number of fine-tuning epochs, and for each checkpoint, we rolled out 1024 episodes from randomized initial states. We predefined the sampled latent code $z$ for each episode (shared across the evaluations of the three seeds) and recorded the resulting environment-defined mode.

The confusion matrices in Figure 14 show that all checkpoints exhibit highly consistent mode assignments across initial-state perturbations; however, the third checkpoint collapses two latent codes into the same mode, mirroring the behavior observed in our 2D Gaussian Mixture analysis, where

**Table 10:** Success rate per mode.

| Mode | C1 ($\uparrow$) | C2 ($\uparrow$) | C3 ($\uparrow$) |
|---|---|---|---|
| 0 | 1.00 | 1.00 | 1.00 |
| 1 | 0.97 | 0.98 | – |
| 2 | 1.00 | 0.91 | 0.95 |
| 3 | 0.97 | 0.89 | 0.92 |

**Table 11:** Pairwise comparison metrics for latent Z stability across checkpoints.

| Pair | NMI ($\uparrow$) | ARI ($\uparrow$) | Z Consistency ($\uparrow$) |
|---|---|---|---|
| C1 vs C2 | 1.00 | 1.00 | 0.00 |
| C1 vs C3 | 0.87 | 0.74 | 0.00 |
| C2 vs C3 | 0.87 | 0.74 | 0.50 |

**Table 12:** Observation noise perturbation.

| Method | SR($\uparrow$) | SR$_\text{M}$ ($\uparrow$) | mc@0.80 ($\uparrow$) | $\mathcal{H}$ ($\uparrow$) |
|---|---|---|---|---|
| PRE | 0.32 | 0.31 | 0.00/4 | 0.99 |
| SP | 1.00 | 0.25 | 1.00/4 | 0.00 |
| SP[MD-MAD] | 0.96 | 0.96 | 4.00/4 | 0.99 |

**Table 13:** Dynamics shift perturbation.

| Method | SR ($\uparrow$) | SR$_\text{M}$ ($\uparrow$) | mc@0.80 ($\uparrow$) | $\mathcal{H}$ ($\uparrow$) |
|---|---|---|---|---|
| PRE | 0.05 | 0.06 | 0.00/4 | 0.90 |
| SP | 0.98 | 0.24 | 1.00/4 | 0.00 |
| SP[MD-MAD] | 0.69 | 0.69 | 2.00/4 | 0.99 |

small state perturbations cause our diversity objective to treat trajectories ending in the same mode as distinct, thereby compressing the latent space. A qualitative visualization of the modes learned by the three checkpoints is shown in Figure 15. We further report the success rate per mode in Table 10.

To quantify seed-level agreement, we compare the checkpoints using three metrics: (i) the Normalized Mutual Information (NMI), which measures the similarity of the mode distributions produced across seeds; (ii) the Adjusted Rand Index (ARI), which evaluates the alignment of the underlying mode-cluster structure; and (iii) a Z-Consistency score, defined as the fraction of latent codes whose dominant mode matches across seeds. As shown in Table 11, NMI and ARI remain high, indicating that all seeds recover consistent sets of behavioral modes, while the collapsed latent in the third checkpoint naturally leads to non-perfect scores. The Z-Consistency score is zero for the first checkpoint comparisons, confirming that although the learned modes are stable, the specific latent-code assignments are not necessarily preserved across seeds, reflecting the inherent permutation symmetry of unsupervised latent discovery.

## I.3 NOISE AND DYNAMICS PERTURBATIONS

This section evaluates the robustness of the proposed method to environmental perturbations beyond reward shifts, specifically focusing on observation noise and dynamics alterations. We conduct these experiments in the *ANYmal* environment. For the observation-noise ablation, we inject Gaussian noise with standard deviation $0.01$ into the observations prior to feeding them into the policy. For the dynamics-shift ablation, we immobilize the first joint of one leg by forcing the corresponding action to zero before each simulator step. These perturbation magnitudes were chosen to avoid completely destabilizing the pre-trained policy: the method assumes a non-trivial degree of multimodality in the underlying model, and more severe interventions (e.g. blocking the second joint) caused immediate falls, thereby collapsing behavioral diversity and falling outside the scope of this study.

We evaluated the steering policy and compared its performance against the same policy trained under the same shifts without the MD-MAD regularization, using the same metrics introduced in the previous section. We further included the performance of the pre-trained model evaluated with the suggested shifts. Taken together, the results in Tables 12 and 13 show that MD-MAD consistently preserves multimodal behavior under observation noise and maintains distinct latent-behavioral modes even when the underlying system dynamics are perturbed. Although dynamic shifts reduce overall task performance, the fine-tuned steering policy still succeeds on two of the four environment modes (per-mode success: (0) 0.02, (1) 0.92, (2) 0.79, (3) 0.98), corresponding to mc@80 = 2/4. This demonstrates robustness of the discovered behavioral modes, while also revealing room for improvement in handling stronger dynamic variations.

**Table 14:** Unique successful action sequences discovered by each method, grouped by number of completed tasks.

| Method | # Tasks | Unique Sequences |
|---|---|---|
| PRE | 1 | [microwave], [kettle] |
| | 2 | [kettle, bottom burner], [microwave, bottom burner], [microwave, kettle] |
| | 3 | - |
| RES / SP / DPPO / DPPO[10] | 1 | [kettle] |
| | 2 | [kettle, bottom burner] |
| | 3 | [kettle, bottom burner, light switch] |
| SP [MD-MAD] | 1 | [microwave], [kettle] |
| | 2 | [microwave, kettle], [kettle, bottom burner] |
| | 3 | [microwave, kettle, bottom burner], [kettle, bottom burner, light switch] |

## I.4 SUCCESSFUL KITCHEN TASKS SEQUENCES

Table 14 reports the distinct successful task sequences executed by each method in the Franka Kitchen environment. The pre-trained policy exhibits multiple valid one- and two-task sequences, while all baselines collapse to a single sequence per task count. In contrast, MD-MAD recovers multiple successful sequences across all levels, preserving most of the original multimodality but losing the sequence "[microwave, bottom burner]".

## I.5 QUALITATIVE VISUALIZATION OF THE LEARNED SKILLS

Figure 16 shows qualitative examples of the trajectory sampled in each environment by the DPPO baseline, as well as the skills learned by the DPPO[MD-MAD] variant trained with our proposed mode discovery and regularization techniques.

# J USE OF LARGE LANGUAGE MODELS (LLMs)

Large Language Models (LLMs) were employed as a general-purpose writing assistant. Specifically, we used LLMs to polish the language, improve readability, and refine the clarity of the manuscript. The models were not used for research ideation, experimental design, data analysis, or interpretation of results. All conceptual contributions, algorithms, experiments, and conclusions presented in this work are solely those of the authors.

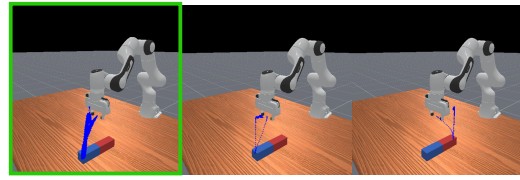

**(a)** *Reach*: DPPO (Left, green box) trajectories and modes learned by DPPO[MD-MAD].

**(b)** *Lift*: DPPO (Left, green box) trajectories and modes learned by DPPO[MD-MAD].

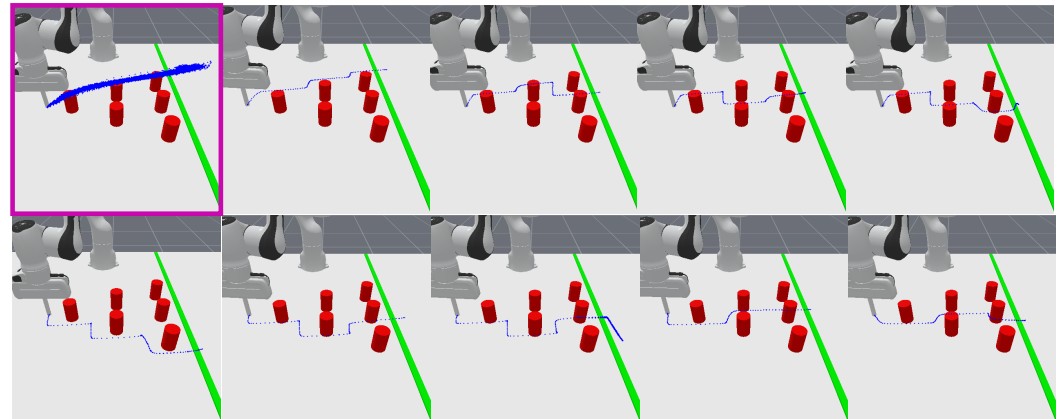

**(c)** *Avoid*: DPPO (Left, purple box) trajectories and modes learned by DPPO[MD-MAD].

**Figure 16:** Visualization of trajectories (blue) from standard fine-tuning and MD-MAD fine-tuning across different tasks. Highlighted boxes (green, purple) show DPPO, which exhibits multimodal behavior only in the *Reach* task. The remaining visualizations represent DPPO[MD-MAD], where trajectories are sampled by varying $z \in \mathcal{Z}$

