# OpenReview forum: "Unsupervised Mode Discovery for Fine-tuning Multimodal Action Distributions"
_ICLR.cc/2026/Conference — Submitted to ICLR 2026_

### Official Review · Reviewer_GJqp · 2025-10-15

**Soundness:** 2
**Presentation:** 1
**Contribution:** 2
**Rating:** 4
**Confidence:** 3

**Summary:**

This paper proposes a method to fine-tune pre-trained generative policies with reinforcement learning while preserving their multimodal behaviours. MD-MAD introduces an unsupervised mode discovery process that identifies latent behavioural modes within pre-trained policies and quantifies multimodality via conditional mutual information. This metric serves as an intrinsic reward during RL. Experiments on robotic manipulation and 2D navigation show that MD-MAD maintains multimodal action distributions and high task performance.

**Strengths:**

1. The paper identifies a clear and underexplored gap, which is a meaningful contribution to the RL + generative modelling community.

2. The figures in this paper is beautiful and informative.

**Weaknesses:**

The main reasons for my hesitation to recommend acceptance are twofold:

The presentation of the method does not clearly convey its motivation or the underlying logic. Even though I am familiar with this research area, I found it difficult to fully grasp the proposed framework. In particular, some passages, for example, lines 194–197 and the use of “therefore” in line 234, lack sufficient explanation to justify the transitions between ideas. I would encourage the authors to improve the exposition of Section 4 by walking the reader step by step through the theoretical reasoning and explicitly clarifying the assumptions and derivations.

I am also concerned about the theoretical formulation of the "multimodal policy". The definition in Question 2 does not appear to align well with standard understandings of multimodality in probability distributions. The authors may want to revisit the conventional definition of a multimodal distribution (e.g., as summarised on Wikipedia, or more rigorously in recent ICLR work [1]) and better explain how their definition connects to or generalises these prior formulations. Establishing this link would make the theoretical foundation of the paper more convincing.

**Questions:**

1. Around line 52, the authors mention that RL fine-tuning often biases the policy towards reward-maximising behaviour at the expense of diversity. I think they are correct and RL fine-tuning *should* bias the policy towards reward-maximising behaviours and ignore sub-optimal modes. Although the authors refer to the issue of reward misalignment, they should further clarify their motivation and explicitly explain why this bias is considered problematic in their setting.

2. The multimodal policy $\pi(a \mid s, w)$ suggests that the action distribution can have multiple peaks. However, the definition in line 182 does not seem to reflect this property. To my understanding, the definition implies that a policy is multimodal if some different input noises induce different action distributions — even if each distribution itself is unimodal. Accordingly, a policy would be unimodal if the action distribution is independent of the input noise. Could the authors confirm whether their notion of multimodality is consistent with previous work, or if it represents a distinct concept?

3. There is a typo in line 207.

4. It is unclear why line 234 states that “distinct values of $z$ can therefore select different behaviours.” After reading Algorithm 1 in the Appendix, my interpretation is that the authors cluster state–action pairs and represent each cluster using the latent code $z$. When PPO is trained, the aim is to control the actor’s behaviour by conditioning on the cluster (source) of the actions. Could the authors confirm whether this interpretation is correct?

**References**

[1] Wang, M., Jin, Y., \& Montana, G. (2025). Learning on One Mode: Addressing Multi-modality in Offline Reinforcement Learning. ICLR.

---

> ### Author Response · Authors · 2025-11-26
> **Addressing Presentation and Methodological Concerns**
>
> We thank the reviewer for their valuable feedback and are especially encouraged by the comments recognizing the novelty of our contribution and the clarity of our figures.
>
> ## Clarity of Presentation and Theoretical Weaknesses
>
> We agree that the original exposition in Section 4 lacked clarity and that our initial definition of multimodality did not align well with standard notions. Following the reviewer’s suggestion and similar feedback from other reviewers, we have substantially revised the method section. In the new version, we removed the suggested definition of a “multimodal policy,” revised our assumptions and theoretical formulations, and explicitly reconnected our notion of multimodality to the conventional view of distributions with multiple modes as discussed in prior work. Moreover, we added the algorithmic description to the main text to visually summarize the full framework. We hope these changes address the reviewer’s concerns regarding both the clarity and the theoretical grounding of our formulation.
>
> ## Questions
>
> [Q1] We agree that RL fine-tuning should bias the policy toward reward-maximizing behavior. The challenge arises when multiple reward-maximizing or near-equivalent behaviors exist: standard RL objectives tend to collapse to one of these solutions as it commits to whichever successful behavior is discovered first, eliminating viable alternative strategies. Our motivation is therefore not to avoid reward maximization, but to avoid loss of valid high-performing modes that are beneficial for robustness, exploration, and downstream controllability. We made this clearer in the revised document.
>
> [Q2] The reviewer’s interpretation is correct. We also acknowledge the concern about our initial formulation of multimodality and have updated the method section by removing the proposed definition.
>
> [Q4] Yes, the reviewer’s interpretation is correct. Algorithm 1 clusters the state–action trajectories induced by the pretrained generative policy, and the latent variable $z \in \mathcal{Z}$  indexes these discovered clusters. During fine-tuning, the steering policy $\pi_\psi^{\mathcal{W}}(w\mid s,z)$ trained with PPO is conditioned on $z$, which allows it to selectively bias the pretrained policy $\pi_\theta(a\mid s,w)$ toward the corresponding mode-specific behavior by steering the sampling of $\mathcal{W}$.
>
> We hope that these clarifications, together with the revisions to the method section and presentation, address the reviewer’s concerns.

---

### Official Review · Reviewer_5s5u · 2025-10-26

**Soundness:** 3
**Presentation:** 3
**Contribution:** 2
**Rating:** 2
**Confidence:** 4

**Summary:**

The authors use mutual information maximization to quantify and promote multimodality of a behavioral policy that is fine-tuned to maximize reward. The framework discovers latent behavior in generative policies that are multimodal. The method is shown to work in some basic environments, and against some baselines.

**Strengths:**

The problem of how to generate multimodal behavior is important. The paper is well written in easy to follow.

**Weaknesses:**

The objective of mutual information maximization does not necessarily promote multimodality. I think there is a general confusion in the paper, as it assumes that higher mutual information between a latent variable and an action is equivalent to multimodality. This is not the case. Therefore, maximizing mutual information does not produce multimodality unless the environment has already several reward modes – as it is the case in the basic example studied.

Treating skills as modes in a latent space of a pre-trained policy has been addressed before in

https://openreview.net/forum?id=HhbHw2yInZ

in a slightly different setting using mixture policies, truly leading with multimodal behaviors. Further discussion and comparison between scopes and methods should be given.

The derivation of the lower bound is not novel, and resembles, and seems equivalent to the once provided in
https://arxiv.org/abs/1907.01657

Therefore, the mathematical derivation is not novel, and no references are provided.

The environments considered are two simple and not high dimensional. For instance, one could use the Ant-v4 environment in a locomotion problem with multiple rewards to test whether the methods produce multimodal behavior in a setting like that of Fig. 4.

**Questions:**

See weaknesses.

---

> ### Author Response · Authors · 2025-11-26
> **Clarifications and Responses**
>
> We sincerely appreciate the reviewer’s thoughtful comments and the recognition that multimodal behavior is an important problem.
>
> We would first like to clarify a possible misunderstanding: the goal of this paper is not to *generate* multimodal behaviors, but to *retain* the multimodal behaviors already present in a pretrained generative policy when applying RL fine-tuning. Existing fine-tuning methods are known to reduce or collapse diversity, and MD-MAD is designed to detect and preserve this pre-existing multimodality. Learning to generate new multimodal behaviors from scratch is a harder problem and outside the scope of this work.
>
> ### **Mutual information and multimodality**
> We agree with the reviewer that maximizing mutual information (MI) between latent $Z$ and actions $A$ doesn’t guarantee multimodality. In our setting, we assume multimodality is already present in the pretrained diffusion policy, and MI is used only to (i) identify latent-consistent modes, and (ii) encourage the steering policy to preserve them under fine-tuning. In practice, the objective we use is $I(Z;S)$ rather than $I(Z;A)$. Taking into account the reviewer’s comments, we revised the method section to remove wording that could suggest MI “produces” multimodality and clarified our maximization objective.
>
> ### **Relation to “Unsupervised Action-Policy Quantization.”**
> We acknowledge the relevance of the suggested reference, which learns a mixture of specialized one-step policies using a max–mix–min entropy criterion, providing task-agnostic action tokens. We highlight that our setting differs in that we do not learn a mixture policy but instead identify and retain latent behavioral modes within a single pretrained generative policy, focusing specifically on diversity preservation during RL fine-tuning and on the latent structure learned by the noise-conditioned class of generative policies. We have added this reference in our extended discussion of related work in Appendix A.1 and commented on the different scopes.
>
> ### **Non-novelty of the lower bound**
> The reviewer is correct that the variational MI lower bound resembles existing formulations [1,2]. We did not claim novelty in the derivation and discussed the connection with prior work in Appendix D.1. To avoid any ambiguity explicitly, we have updated the method section to cite prior work more clearly as suggested by the reviewer.
>
> ### **Locomotion environment**
> We appreciate the suggestion to include a locomotion task analogous to Ant-v4. In the revised version, we added two new evaluation settings, including a high-dimensional locomotion task with multiple reward modes, executed with an ANYmal quadruped robot. We chose the ANYmal against the Ant-v4 for ease of integration within our ManiSkill framework. The results shown in  Table 5 of the updated document confirm the findings reported in the original submission, including improved mode retention and stable multimodal rollouts. We also provide a visualization analogous to Figure 4 for the new environment in Appendix I.2 (Figure 15), as suggested by the reviewer.
>
> We hope that these clarifications and the extended experimental evaluation adequately address the reviewer’s concerns.
>
> [1] Eysenbach, Benjamin, et al. "Diversity is all you need: Learning skills without a reward function." arXiv preprint arXiv:1802.06070 (2018).
>
> [2] Sharma, A., Gu, S., Levine, S., Kumar, V. and Hausman, K., 2019. Dynamics-aware unsupervised discovery of skills. arXiv preprint arXiv:1907.01657.

---

### Official Review · Reviewer_5sqb · 2025-10-30

**Soundness:** 4
**Presentation:** 3
**Contribution:** 3
**Rating:** 6
**Confidence:** 4

**Summary:**

The paper proposes MD–MAD, a framework for fine‑tuning pre‑trained diffusion (or flow) policies with RL while explicitly preserving multimodality. Core ideas: (i) formalize multimodality via conditional mutual information; (ii) discover latent behavioral modes by reparameterizing a steering policy with a latent variable z and training an inference model $q_\phi(z\mid s, a)$; (iii) use the resulting MI lower bound as an intrinsic reward to regularize RL fine‑tuning.

Experiments on a 2D Gaussian‑mixture landscape and robotic manipulation (Reach, Lift, Avoid) show improved task success and mode retention versus DPPO, residual policies, and steering baselines.

**Strengths:**

1. On methodology: First, multi-modality MI is practical via a variational lower bound and a latent-conditioned steering policy. Second, such intrinsic MI rewards integrate cleanly with PPO and are agnostic to the RL algorithm. Third, presentations on formulation and motivation are clear.

2. On Experiments: MD–MAD variants recover full mode coverage under balanced and unbalanced rewards (Table 2, p. 8). On ManiSkill/D3IL tasks, it retains diversity (SR_M, mc@0.8, entropy) with minimal loss in SR, often improving it (Tables 3–4, p. 9).

Also, the appendix clearly situates itself among direct fine‑tuning / residual / steering strategies and shows MD–MAD is orthogonal and can augment them.

**Weaknesses:**

1. Method:
(1) The theoretical rationale is mostly heuristic; no formal guarantees are provided for convergence or z identifiability.
(2) Introducing z as the latent factor governing w is risky; without additional structure or constraints, z could become arbitrary or trivial, making the reason it works unclear.
(3) Training and maintaining both the steering and inference networks add complexity and hyperparameter sensitivity.

2. Experiments:
(1) Stress tests mainly vary reward landscapes (rotations/imbalance) and task types; there is little evaluation under dynamics shifts, sensor noise, or partial observability, limiting robustness claims.
(2) The paper uses MI estimates and entropy-based indicators but lacks more direct trajectory diversity metrics.
(3) I think the [mode discovery + MI] backbone should include the unsupervised RL baselines,  VALOR [1], DIAYN [2], and Controllability-Aware Skill Discovery [3].

I did not check the details in the appendix due to limited review time.

[1] Achiam, Joshua, et al. "Variational option discovery algorithms." arXiv preprint arXiv:1807.10299 (2018).

[2] Eysenbach, Benjamin, et al. "Diversity is all you need: Learning skills without a reward function." arXiv preprint arXiv:1802.06070 (2018).

[3] Park, Seohong, et al. "Controllability-aware unsupervised skill discovery." arXiv preprint arXiv:2302.05103 (2023).

**Questions:**

1. Is the latent stable across seeds/perturbations (e.g., confusion matrices over modes, NMI/ARI, per-mode SR)?

2. How does MD–MAD behave under dynamic changes or observation noise? Any evidence that the MI estimator remains stable when the state distribution drifts during fine‑tuning?

3. Could you add an independence/disentanglement constraint to improve the controllability of z (e.g., KL regularization, TC-style regularization)? (Re. Weak 1 (2))

---

> ### Author Response · Authors · 2025-11-26
> **Part 1 - Weaknesses**
>
> We thank the reviewer for their positive evaluation of our methodology and experimental results, and we appreciate the helpful and constructive feedback.
>
> ## Weaknesses
>
> **[W1.1]** Regarding theoretical guarantees, we agree that our approach, like most variational MI-based skill-discovery methods such as DIAYN [1], VALOR [2], and related work, does not provide formal convergence guarantees or identifiability results.
>
> **[W1.2]** On the concern that introducing a latent $\mathcal{Z}$ to govern the noise $\mathcal{W}$ may lead to degeneracies, we agree that this is a potential risk. In practice, using a categorical $\mathcal{Z}$ helps avoid trivial reconstructions, since the continuous latent space $\mathcal{W}$ of the diffusion policy already contains rich structure under the multimodality assumption. Collapse of certain z-modes to the same behavior can still occur, partly due to known limitations of MI-based diversity objectives. While the main objective of this work is primarily to evaluate the feasibility of using diversity regularization to preserve multimodality during fine-tuning, exploring alternative regularization objectives, as the reviewer suggests in Question #3, is an important direction that we highlight for future work.
>
> **[W1.3]** We also acknowledge that jointly training a steering and an inference model introduces additional complexity. However, we found training to be significantly more stable than in standard skill-discovery methods because exploration is fully determined by the pretrained generative policy, rather than being learned from scratch.
>
> **[W2.1 -2.2]** We expanded the evaluation section to include stress tests over dynamics shifts and sensor noise in Appendix I.2 (See answers to Question 2 for more details).
>
> Regarding the lack of direct trajectory diversity metrics and success rate per mode, while we do not report the raw success rate $SR_{i}$ for each individual mode, our metrics
> $\mathrm{SR}_{\mathrm{M}} = \tfrac{1}{K}\sum_i \mathrm{SR}_i$ and $\mathrm{mc}\@\tau$
> are explicit aggregations over per-mode success rates and thus reflect mode-level performance rather than only global success. In particular, $\mathrm{mc}@\tau$ directly encodes whether each mode reaches a sufficiently high success level.
>
> We agree that trajectory-diversity measures such as Dynamic Time Warping (DTW) could, in principle, provide additional insights. In practice, we found DTW to be highly sensitive to outliers, which made it difficult to interpret. Given that we have simulator-defined ground-truth modes, our diversity metrics *mode entropy* and *mode coverage* already reflect trajectory-level variability: in our setup, two trajectories are assigned to different ground-truth modes when they differ in behavior despite reaching the same goal. Thus, mode entropy in this case provides a proxy for trajectory diversity.
>
> **[W2.3]** We would like to clarify that our backbone is essentially DIAYN [1] but integrated within a steering-policy reparameterization to leverage the pre-trained generative policy. The pre-trained policy provides meaningful state-space coverage, avoiding the exploration difficulties common in skill-discovery methods. As noted by the reviewer, extending MD-MAD with more advanced skill-discovery algorithms such as CSF[3], METRA[4], or controllability-aware skill discovery[5] is a compelling direction that we plan to pursue. As mentioned in the answer to [W1.2], the objective of this work was to establish a middle ground between unsupervised skill discovery and RL fine-tuning for multimodal generative policies. Further exploring the extension of the mode discovery component is an exciting future work.
>
> [1] Eysenbach, Benjamin, et al. "Diversity is all you need: Learning skills without a reward function." arXiv preprint arXiv:1802.06070 (2018).
>
> [2] Achiam, Joshua, Harrison Edwards, Dario Amodei, and Pieter Abbeel. "Variational option discovery algorithms." arXiv preprint arXiv:1807.10299 (2018).
>
> [3] Zheng, C., Tuyls, J., Peng, J. and Eysenbach, B. Can a MISL fly? analysis and ingredients for mutual information skill learning. ICLR 2025.
>
> [4] Park, S., Rybkin, O. and Levine, S., 2023. Metra: Scalable unsupervised rl with metric-aware abstraction. ICLR 2024.
>
> [5] Park, Seohong, et al. "Controllability-aware unsupervised skill discovery." arXiv preprint arXiv:2302.05103 (2023).

---

> ### Author Response · Authors · 2025-11-26
> **Part 2 - Questions**
>
> ## Questions
>
> **[Q1]** We conducted additional analyses to assess latent stability across seeds and under state perturbations. We included confusion matrices, NMI/ARI scores, and per-mode SR in Appendix I.2. We observed that, for a given trained policy, the mapping of $z$ to behavioral mode is consistent under state perturbations, while permutations across seeds do occur. We note that cross-seed alignment is not an objective of MD-MAD: our goal is to sample trajectories from the pretrained policy, cluster them into discriminable latent categories, and reliably condition on these categories during fine-tuning to preserve multimodality. Because the latent categories are exchangeable up to permutation, what matters is internal consistency rather than cross-seed naming.
>
> **[Q2]** We integrated robustness tests under mild dynamics shifts and observation noise, and report results in Appendix I.3. Preliminary results indicate that MD-MAD maintains mode separability under moderate shifts, but, consistent with the reviewer’s intuition, we would expect the performance to deteriorate if the underlying pretrained policy loses its multimodality due to a severe distribution shift, as the inference model would no longer be able to discriminate modes.
>
> **[Q3]** In our current setting, $\mathcal{Z}$ is a single categorical latent indexing discrete behavioral modes, so there is no internal factorization over dimensions on which to impose a TC-style independence constraint. However, we fully agree that exploring continuous latent parameterizations, where disentanglement regularizers could meaningfully shape latent geometry and further enhance controllability, is a promising direction that we highlight in Section 6 for future work.
>
> We hope the provided clarifications and revisions help resolve the reviewer’s concerns.

---

### Official Review · Reviewer_MrJX · 2025-11-03

**Soundness:** 2
**Presentation:** 2
**Contribution:** 2
**Rating:** 4
**Confidence:** 4

**Summary:**

In this work, the authors present MD-MAD, an unsupervised discovery algorithm for finding modes of multimodal action policies. This algorithm tries to solve the problem of loss of multimodality when fine-tuning a multimodal behavior policy. The way the authors address this is by trying to associate a latent-conditioning behavior eliciting model, which given the conditioning would generate noise that when passed into a policy will create the actions from the right modes. The authors train this through a mutual information maximization lens, by optimizing for the mutual information between the latent conditioning and the generated actions.

The authors show experiments in two domains: one in a simple gaussian mixture environment, and another set on simulated ManiSkill environments. In the gaussian setup, the authors show that they are able to extract the modes. In the simulation setup, the experiments show that adding MD-MAD as a regularization maintains higher multimodality.

**Strengths:**

1. This work finds a shared middle ground between unsupervised skill discovery work and multimodal imitation learning. The connection is quite interesting, since the skill extraction behaves similarly in both cases, and the similarity of algorithms is quite insightful.
2. MD-MAD is compatible with diverse fine-tuning paradigms, such as policy decorator, residual learning, or DPPO. This property is quite helpful in terms of being useful to practitioners.

**Weaknesses:**

1. The simulated environment benchmarking are only on three tasks, and the extent to which multimodality exists in those tasks is not clear in the first place. A better way to evaluate the multimodality of trained policies would be to use something like the Franka Kitchen environment where there is clear distributional multimodality in behavior and the results are much more easily quantifiable.
2. Without any real robot experiments, it is hard to tell whether this would scale in real, and what kind of bottlenecks could be there in the process.
3. Similar to unsupervised skill discovery, finding the right parametrization for the latent space Z seems challenging for large real world datasets, and merits more discussion.
4. The jump from single-action multimodality to trajectory level multimodality is not clear – given that policies can vary or multiplex similar Zs for different actions at different subsets of the state space.

**Questions:**

1. The algorithm currently uses a two stage method, but with some human labels or demonstrations, would it be possible to do this in a single stage?
2. What are the challenges towards stability that the authors see in this work? Unsupervised skill discovery never took off due to stability problems in real application, so what are the primary stabilization approaches that can benefit MD-MAD?

---

> ### Author Response · Authors · 2025-11-26
> **Part 1 - Weaknesses**
>
> We thank the reviewer for the constructive and detailed feedback. We are glad that the reviewer found the connection between unsupervised skill discovery and multimodal imitation learning insightful and appreciated MD-MAD’s compatibility with diverse fine-tuning paradigms.
>
> ## Weaknesses
>
> **[W1]** We agree that the original set of evaluation tasks was limited in scope. Following the reviewer’s suggestion, we first clarified the nature of multimodality in our existing task descriptions in Appendix H. We further expanded the experiments to include the (suggested) Franka Kitchen environment, where trajectories exhibit a clear distributional diversity, and added a high-dimensional locomotion task to test scalability. In both new settings, MD-MAD consistently helped preserve the original multimodality of the pre-trained policy. The corresponding results are presented in Section 5.3, specifically in Tables 5 and 6 for the locomotion and Franka Kitchen environments, respectively.
>
> **[W2]**  We identify two main challenges for real-world deployment of MD-MAD: (i) the sim-to-real gap, and (ii) scaling the framework to vision-based policies. Because MD-MAD relies on online RL fine-tuning, it is most naturally trained in simulation since online RL remains expensive for physical robots. This requires addressing the sim-to-real gap for real-world deployment. A practical path forward to address this challenge is to rely on recent progress in 3D state estimation, such as FoundationPose[1], which could provide sufficiently accurate object and pose estimates to deploy MD-MAD with minimal modifications to the core method. In parallel, we see a promising opportunity to extend MD-MAD to operate directly from visual observations. This would likely require adapting the mode-discovery module, drawing inspiration from recent work in skill discovery that incorporates visual or latent perceptual representations [2]. We consider both directions, sim-to-real transfer and visual-based extensions, to be exciting avenues for future work.
>
>
> **[W3]** The reviewer raises an important point regarding the parametrization of the latent space $\mathcal{Z}$ for large and potentially heterogeneous datasets. We see two key considerations in this setting: (i) learning from multi-task datasets, and (ii) choosing an appropriate latent parametrization (discrete vs. continuous, dimensionality).
>
> First, when scaling to large multi-task datasets, the question becomes whether the latent space should capture dataset-level diversity or task-specific behavioral modes. MD-MAD is designed as an action-level mode extractor for task-conditioned policies: many real-world datasets include task labels, but not intra-task behavioral diversity. Our framework focuses on discovering and preserving these within-task modes. Extending MD-MAD to recover hierarchical structure (e.g., dataset-level task clusters and task-level behavioral modes) would likely require a hierarchical latent space, an exciting direction we now explicitly acknowledge in Section 6.
>
> Second, independently of hierarchy, large datasets also raise the question of latent parametrization. Our experiments with discrete latent spaces suggest that mild overparameterization is not problematic: as shown in our Gaussian example, MD-MAD can reliably collapse redundant codes and recover the relevant modes. Thus, for discrete codes, a practical strategy is to allocate a sufficiently large latent space and let MD-MAD identify the effective subset. Continuous latent spaces, possibly paired with regularization as in recent skill-discovery works [2,3], offer the potential to represent unbounded behavioral diversity and may further improve expressivity. Exploring such continuous or hybrid parametrizations within MD-MAD constitutes promising future work.
>
> We have integrated this extended discussion into the revised Section 6 of the paper.
>
> **[W4]** Concerning the transition from single-action multimodality to trajectory-level multimodality, we clarified in the text that our assumption flows from trajectory-level multimodality to action-level multimodality, not the other way around. The previous explanation incorrectly suggested an implication in the opposite direction, and we have updated Section 4 to avoid this confusion.
>
> [1] Wen, B., Yang, W., Kautz, J. and Birchfield, S., 2024. Foundationpose: Unified 6d pose estimation and tracking of novel objects. In Proceedings of the IEEE/CVF Conference on Computer Vision and Pattern Recognition (pp. 17868-17879).
>
> [2] Park, S., Rybkin, O. and Levine, S., 2023. Metra: Scalable unsupervised rl with metric-aware abstraction. ICLR 2024.
>
> [3] Zheng, C., Tuyls, J., Peng, J. and Eysenbach, B. Can a MISL fly? analysis and ingredients for mutual information skill learning. ICLR 2025.

---

> ### Author Response · Authors · 2025-11-26
> **Part 2 - Questions**
>
> ## Questions
>
> **[Q1]** Regarding the two-stage training pipeline, we note that incorporating labels could help guide the learning, but would not eliminate the need for a two-stage setting. Even with labels, the inference model must be pre-trained to discriminate different modes to provide an informative mutual information (MI) estimate during fine-tuning. Without this pre-training, the policy could change faster than the inference model, causing modes to collapse before the model has learned to recognize the original modes. Labels could nonetheless be useful in reducing joint-training instabilities, for example, by allowing supervised training of the inference model.
>
> **[Q2]** Finally, regarding stability and its relation to unsupervised skill-discovery methods, we observe improved stability in MD-MAD because the pretrained policy already provides meaningful state-space coverage, avoiding the exploration difficulties that skill-discovery methods face when learning from scratch. In our first phase, the agent only manipulates the latent noise of a diffusion policy, so exploration is predetermined by the pre-trained policy. During RL fine-tuning, some instability remains due to shifts in the state distribution, which requires balancing updates between the steering policy and the inference model. In practice, alternating their updates was sufficient to maintain stability.  As discussed for Q1, incorporating task labels when available could help stabilize the training.
>
> We hope that the clarifications and additional analyses provided above address the reviewer’s concerns.

---

### Author Response · Authors · 2025-12-02
**Discussion Summary (1/3)**

Dear PCs, SACs, ACs, and Reviewers,

We would like to begin by sincerely thanking the reviewers for their careful, thoughtful, and constructive feedback. Their insights substantially improved the clarity and quality of the revised manuscript. To assist the newly assigned Area Chair, we provide below a summary of the strengths highlighted in the reviews and the concrete steps we took to address the identified weaknesses and concerns.
## What we Propose
*Mode Discovery for Multimodal Action Distributions* (MD-MAD), a method for preserving the multimodal behaviors encoded in pre-trained generative policies during reinforcement learning (RL) fine-tuning.
## Strengths
We sincerely appreciate and are encouraged by the positive aspects highlighted in the initial reviews. In particular, reviewers noted:
- **Conceptual contribution:** The connection drawn between unsupervised skill discovery and multimodal fine-tuning of generative policies is interesting and quite insightful (MrJX: strengths).
- **Practical relevance:** MD–MAD, is broadly applicable across fine-tuning paradigms, which makes it useful to practitioners (MrJX: strengths, 5sqb: strengths); moreover, the mutual-information intrinsic reward is practical and agnostic to the underlying RL algorithm (5sqb: strengths).
- **Quality of presentation:** The motivation and exposition were found to be clear (5sqb: strengths, 5s5u: strengths), and the figures and visualizations were described as particularly clear and informative (5sqb: strengths).
- **Importance of the problem:** The work targets an important and underexplored gap (5s5u: strengths, GJqp: strengths), which is a meaningful contribution to the RL + generative modelling community (GJqp: strengths).
- **Empirical results:** The experiments demonstrate that MD–MAD retains the multimodal structure of the pre-trained policy during fine-tuning, achieving comparable or improved task success compared to baselines that collapse to a single mode (5sqb: strengths).

---

> ### Author Response · Authors · 2025-12-02
> **Discussion Summary (2/3)**
>
> ## Methodology, Formalism, and Presentation
> Below, we summarize the main methodological and presentation-related concerns raised by the reviewers, along with the corresponding revisions made in response.
> - (MrJX: Weakness 3, 5sqb: Weakness 1.2 and Question 3) **Finding the correct parameterization of the latent space $\mathcal{Z}$ is challenging and warrants further discussion.**
>    - We expanded the discussion of latent-space parametrization (Section 6) to note that discrete latent spaces are practical in our setting: mild overparameterization reliably collapses redundant codes and recovers the relevant modes, making the choice of $ |\mathcal{Z}|$ non-critical. We further highlight continuous or hybrid latent parametrizations, potentially combined with disentanglement or controllability-aware regularization, as a natural and promising direction for future work.
> - (MrJX: Question 2, 5sqb: Weakness 1.3) **Unsupervised skill discovery tends to have stability problems, and training and maintaining both the steering and inference networks add complexity and hyperparameter sensitivity.**
>    - Although MD–MAD introduces both a steering and an inference network, it is substantially more stable than standard skill-discovery methods because it does not learn skills from scratch. The pre-trained generative policy fixes the exploration distribution, avoiding the instability and hyperparameter sensitivity typically associated with learning new behaviors.
> - (5s5u: Weakness 1, GJqp: Weakness 2) **The objective of mutual information maximization does not necessarily promote multimodality, and the definition of "multimodal policy" is not standard.**
>    - We clarified that multimodality is assumed to be already present in the pre-trained policy; MI is used only to identify mode-consistent latents and preserve them during fine-tuning. Practically, our objective maximizes $I(Z;S)$ rather than $I(Z;A)$. We revised the method section to avoid any implication that MI itself induces multimodality.
>    - We revised the assumptions and theoretical formulations in Section 4 and explicitly reconnected our notion of multimodality to the standard definition of distributions with multiple modes, as established in prior work.
> - (5s5u: Weakness 2,3) **Non-Novelty of the method and the variational lower bound.**
>    - We acknowledge the relevance of the reviewer’s suggested reference and clarified that the cited work focuses on learning new skills from scratch, whereas MD–MAD aims to retain the modes already present in a pre-trained generative policy.
>    - While the connection to prior work on variational mutual-information lower bounds was already discussed in Appendix D.1, we further revised the method section to cite these formulations explicitly and avoid any ambiguity regarding novelty.
> - (GJqp: Weakness 1 and Question 1, MrJX: Weakness 4) **The presentation of the method does not clearly convey its motivation or the underlying logic, and the connection from action-level multimodality to trajectory-level multimodality is unclear.**
>    - We restructured the exposition of Section 4 to present the assumptions and derivations in a step-by-step manner and added an algorithmic summary of MD–MAD in the main text. We also made the motivation for retaining multiple high-reward behaviors (robustness, controllability) more explicit.
>    - We now explicitly state that our assumption proceeds from trajectory-level multimodality to action-level multimodality, rather than the reverse, and revised Section 4 accordingly to eliminate this source of confusion.

---

> ### Author Response · Authors · 2025-12-02
> **Discussion Summary (3/3)**
>
> ## Experiments and Evaluation
> Below, we summarize the main experimental and evaluation concerns raised by the reviewers and outline the corresponding revisions implemented in the updated manuscript.
> - (MrJX: Weakness 1, 5s5u: Weakness 4) **The simulated environment benchmarking is limited.**
>    - We expanded the empirical evaluation beyond the original set of tasks to incorporate the settings suggested by the reviewers: (i) a high-dimensional quadruped locomotion task (ANYmal) exhibiting multiple reward modes, and (ii) the Franka Kitchen environment, where trajectory-level multimodality is well defined and easily quantifiable. The new results (Section 5.3, Tables 5–6 and associated figures) demonstrate that MD–MAD consistently preserves mode diversity while maintaining task success, even in these more challenging scenarios.
> - (5sqb: Weakness 2.1 and Question 2) **Stress tests only vary reward landscapes.**
>    - We extended the evaluation to include robustness tests under dynamics shifts and sensor noise, reported in Appendix I.2. The results show that MD–MAD maintains mode separability under moderate perturbations. However, performance would degrade under severe distribution shifts that compromise the multimodality of the underlying pre-trained policy itself, which lies outside the scope of the present work.
> - (5sqb: Weakness 2.2 and Question 2) **The papers lack trajectory diversity metrics and per-mode success rate.**
>    - We clarified that our evaluation metrics explicitly aggregate performance across modes (rather than only reporting global success) and inherently capture trajectory diversity through the simulator-defined ground-truth modes. These metrics, therefore, reflect both per-mode success and trajectory diversity.
> - (5sqb: Weakness 2.3) **The [mode discovery + MI] backbone should include the unsupervised RL baselines.**
>    - We clarified that the backbone of our approach is essentially DIAYN [1], adapted through a steering-policy reparameterization that enables integration with a pre-trained generative policy. This required a modified architecture and methodology to make DIAYN-style objectives compatible with the structure and constraints of diffusion- and flow-based policies. We also emphasized in Section 6 that incorporating more advanced skill-discovery methods into MD–MAD is a natural and promising direction for future work.
> -  (5sqb: Question 1) **Is the latent $\mathcal{Z}$ stable across seeds/perturbations?**
>    - We added analyses of latent stability across seeds and state perturbations, including confusion matrices, NMI/ARI scores, and per-mode success rates (Appendix I.2). For a given trained policy, the mapping from $z$ to behavioral modes is stable under state perturbations, though permutations across seeds naturally occur. Cross-seed alignment is not an objective of MD–MAD; the goal is simply to obtain consistent latent categories for preserving multimodality during fine-tuning.
>
>
> [1] Eysenbach, Benjamin, et al. "Diversity is all you need: Learning skills without a reward function." arXiv preprint arXiv:1802.06070 (2018).
>
>
> ---
>
> We hope that this overview helps contextualize the reviewers’ assessments and clearly conveys how their feedback directly informed the revisions made to the manuscript.
>
> Sincerely,
>
> The Authors

---

### Meta-Review · Area_Chair_bB3B · 2026-01-12

**Summary:**

- The key theme of the reviewer's concerns were around the theoretical framework of the proposed multimodality measure and the breadth of empirical validation.
- Reviewers MrJX and 5s5u initially argued that the 2D Gaussian and ManiSkill tasks were insufficient to prove the method's scalability. During the rebuttal period,the authors added evaluation on the Franka Kitchen and ANYmal envs.
- Reviewers pointed out several clarifications and confusions in the writing about the meaning of multi-modality in policies in the typical RL literature. The authors did take steps towards this but the paper does need to be framed better for clarity.
- There were concerns about the stability of unsupervised skill discovery and the sensitivity of the latent dimensionality. Authors reported new stability analysis oriented experiments and experiments to ablate latent dimensionality.
- This paper is indeed studying an interesting problem and it will become more relevant as foundation models become more prevalent as the reusable torso in deep reinforcement learning. But the reviewers could not reach clear consensus due to various ongoing or unresolved issues / limitations.

**Reviewer Concerns:**

Addressed:
- Reviewers MrJX and 5s5u initially criticized the limited scope of the 2D and ManiSkill tasks. In response, the authors partially addressed this by adding evaluation on ANYmal locomotion and Franka Kitchen.
- Reviewers GJqp and 5s5u were concerned that the paper implied Mutual Information (MI) creates multimodality. The authors clarified that multimodality is assumed to be pre-existing from demonstrations. However, the paper needs to be framed and written with significantly better clarity.

Unresolved:
- Reviewer MrJX noted the lack of real robot experiments. The authors acknowledged that MD-MAD currently relies on simulated RL fine-tuning due to the high sample cost.
- As noted by Reviewer 5sqb, the method remains a heuristic variational approach. While empirically successful, there are no formal proofs for the convergence. This puts more weights on the scope and diversity of experimental validation.
- The current framework uses low-dim states. Expanding it to vision based policies remains open and is important given most multi-modal pre-trained models will be multimodal in the near future.

**Reviewer Scores:**

- MrJX: This reviewer's main concern was the limited scope of the simulated environments and there was some updates from the authors to mitigate this.

- 5sqb: Requested stress tests beyond reward landscapes and more direct metrics. The authors provided new robustness tests against dynamics shifts and sensor noise.

- 5s5u: This reviewer gave the lowest score due to a perceived confusion between mutual information and multimodality and the lack of a high dimensional locomotion task like Ant-v4. The authors did address these concerns to some extent and reviewer could have increased their scores.

- GJqp: Similar clarity, formulation and writing concerns as above.

To summarize, the shared issues were: lack of real world and diverse experiments, hyper parameter sensitivity, and lack of a robust formal framework / analysis.

---

### Decision · Program_Chairs · 2026-01-26

Reject